# Blumenols as shoot markers of root symbiosis with arbuscular mycorrhizal fungi

**Ming Wang[1†], Martin Schäfer[1†‡], Dapeng Li[1], Rayko Halitschke[1], Chuanfu Dong[2§], Erica McGale[1], Christian Paetz[3], Yuanyuan Song[1#], Suhua Li[1], Junfu Dong[1,4], Sven Heiling[1¶**], Karin Groten[1], Philipp Franken[5,6], Michael Bitterlich[5], Maria J Harrison[7], Uta Paszkowski[8], Ian T Baldwin[1*]**

[1]Department of Molecular Ecology, Max Planck Institute for Chemical Ecology, Jena, Germany; [2]Department of Bioorganic Chemistry, Max Planck Institute for Chemical Ecology, Jena, Germany; [3]Research Group Biosynthesis / NMR, Max Planck Institute for Chemical Ecology, Jena, Germany; [4]College of Life Sciences, University of Chinese Academy of Sciences, Beijing, China; [5]Leibniz-Institute of Vegetable and Ornamental Crops, Grossbeeren, Germany; [6]Institute of Biology, Humboldt Universität zu Berlin, Berlin, Germany; [7]Boyce Thompson Institute for Plant Research, Ithaca, United States; [8]Department of Plant Sciences, University of Cambridge, Cambridge, United Kingdom

*For correspondence:
baldwin@ice.mpg.de

†These authors contributed equally to this work

Present address: ‡Institute for Evolution and Biodiversity, University of Münster, Münster, Germany; §Department of Integrative Evolutionary Biology, Max Planck Institute for Developmental Biology, Tübingen, Germany; #College of Crop Science, Fujian Agriculture and Forestry University, Fuzhou, China; ¶Institute of Clinical Chemistry and Laboratory Diagnostics, Jena University Hospital, Jena, Germany; **Integrated Biobank Jena, Jena University Hospital, Jena, Germany

**Abstract** High-through-put (HTP) screening for functional arbuscular mycorrhizal fungi (AMF)-associations is challenging because roots must be excavated and colonization evaluated by transcript analysis or microscopy. Here we show that specific leaf-metabolites provide broadly applicable accurate proxies of these associations, suitable for HTP-screens. With a combination of untargeted and targeted metabolomics, we show that shoot accumulations of hydroxy- and carboxyblumenol C-glucosides mirror root AMF-colonization in *Nicotiana attenuata* plants. Genetic/ pharmacologic manipulations indicate that these AMF-indicative foliar blumenols are synthesized and transported from roots to shoots. These blumenol-derived foliar markers, found in many di- and monocotyledonous crop and model plants (*Solanum lycopersicum, Solanum tuberosum, Hordeum vulgare, Triticum aestivum, Medicago truncatula* and *Brachypodium distachyon*), are not restricted to particular plant-AMF interactions, and are shown to be applicable for field-based QTL mapping of AMF-related genes.
DOI: https://doi.org/10.7554/eLife.37093.001

## Introduction

More than 70% of all higher plants, including crop plants, form symbiotic associations with arbuscular mycorrhizal fungi (AMF) (*Brundrett and Tedersoo, 2018*). While the fungus facilitates the uptake of mineral nutrients, in particular phosphorous (P) and nitrogen, the plant supplies the fungus with carbon (*Helber et al., 2011*; *Bravo et al., 2017*; *Jiang et al., 2017*; *Keymer et al., 2017*; *Luginbuehl et al., 2017*). The interaction affects plant growth (*Rooney et al., 2009*; *Adolfsson et al., 2015*) and resistance to various abiotic and biotic stresses (*Pineda et al., 2010*; *Vannette et al., 2013*; *Chitarra et al., 2016*; *Sharma et al., 2017*). Although AMF interactions are physically restricted to the roots, they influence whole-plant performance, hence systemic metabolic responses have been anticipated, and searched for, but no general AMF-specific responses have been found (*Bi et al., 2007*; *Toussaint, 2007*; *Schweiger and Müller, 2015*). While changes in foliar levels of carbohydrates, proteins, and amino acids, as well as secondary metabolites and

**eLife digest** All plants need a nutrient called phosphorus to grow and thrive. Phosphorus is found in soil, but the supply is limited so plants often struggle to acquire enough of it. To overcome this problem, many plants form friendly relationships (or symbioses) with certain fungi in the soil known as arbuscular mycorrhizal fungi. The fungi colonize plant roots and supply phosphorus and other nutrients in return for sugars and various molecules.

Although many crop plants – including barley and potatoes – are able to form these symbioses, farmers commonly apply fertilizers containing phosphate and other nutrients to their fields to increase the amount of food they produce. Breeding new crop varieties that are better at forming symbioses with the fungi could reduce the need for fertilizers. However, the methods currently available to study these relationships are laborious and time-consuming, typically requiring samples of plant roots to be examined in a laboratory.

Wang, Schäfer et al. used an approach called metabolomics to search for molecules in coyote tobacco plants that indicate the plants have formed symbioses with arbuscular mycorrhizal fungi. The experiments found that a group of molecules called blumenols accumulate in the roots and also in the shoots and leaves of plants with these symbioses, but not in the tobacco plants that were not able to associate with the fungi. Experiments in several other plant species including tomato, potato and barley produced similar findings, suggesting that the blumenols may be a useful and potentially universal indicator of symbioses between many different plants and fungi.

Measuring the levels of blumenols in plant shoots and leaves is much quicker and easier than current methods of identifying fungal symbioses in plant root samples. Therefore, blumenols may be a useful tool for plant breeders who would like to screen large numbers of plants for these symbioses, and breed crops that negotiate better interactions with the beneficial fungi.

DOI: https://doi.org/10.7554/eLife.37093.002

phytohormones have been shown to respond to AMF inoculation (*Schweiger et al., 2014*; *Aliferis et al., 2015*; *Adolfsson et al., 2017*), these changes are not specific to AMF interactions and tend to be general responses to various abiotic and biotic stresses. Moreover, these metabolic responses also tend to be taxa-specific, and many are likely indirect consequences of AMF-mediated effects on plant growth and development.

In contrast, large amounts of blumenol-type metabolites accumulate in roots after AMF inoculation. These compounds are apocarotenoids, in particular $C_{13}$ cyclohexenone derivatives, produced by the cleavage of carotenoids. After AMF colonization, a $C_{40}$ carotenoid is cleaved by carotenoid cleavage dioxygenase 7 (CCD7) to produce a $C_{13}$ cyclohexenone and a $C_{27}$ apocarotenoid which is further cleaved by CCD1 to yield a second $C_{13}$ cyclohexenone (*Floss et al., 2008*; *Vogel et al., 2010*; *Hou et al., 2016*). The compounds have been found to accumulate in the roots of AMF-colonized plants in a manner highly correlated with the fungal colonization rate (*Fester et al., 1999*). Other stimuli such as pathogen infection and abiotic stresses, do not induce their accumulations (*Maier et al., 1997*). The AMF-induced accumulation of these compounds is widespread and has been observed in roots of plant species from different families, including mono- and dicotyledons, (*Hordeum vulgare*, *Peipp et al., 1997*); *Solanum lycopersicum* and *Nicotiana tabacum*, *Maier et al., 2000*; e.g., *Zea mays*, *Fester et al., 2002*; *Lotus japonicus* and *Medicago truncatula*, *Fester et al., 2005*; *Ornithogalum umbellatum*, *Schliemann et al., 2006*; *Allium porrum*, *Schliemann et al., 2008*).

Blumenols are classified into three major types; blumenol A, blumenol B and blumenol C (*Figure 1A*). However, previous studies have reported that only blumenol glycosides containing a blumenol C-based aglycone are positively correlated with mycorrhizal colonization. The aglycone can be additionally hydroxylated at the C11 or carboxylated at the C11 or C12 position (*Maier et al., 1997*; *Maier et al., 2000*). Additionally, 7,8-didehydro versions of blumenol C have been reported (*Peipp et al., 1997*). The glycosylation usually occurs as an *O*-glycoside at the C9 position (*Strack and Fester, 2006*), but glycosylations at the hydroxylated C11 position have also been observed (*Schliemann et al., 2008*). The glycosyl moiety can be a single sugar or combinations of glucose (Glc), rhamnose, apiose, arabinose and/or glucuronic acid, which, in turn can be

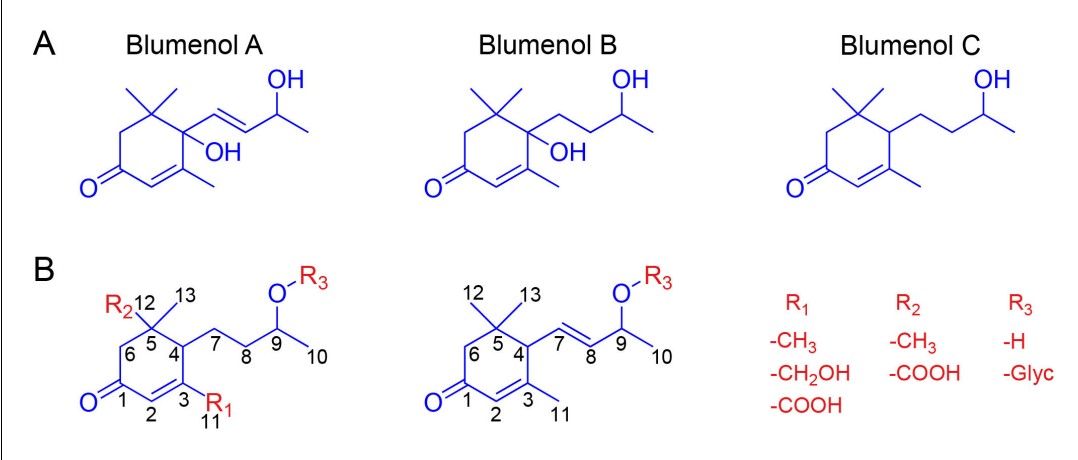

**Figure 1.** Blumenol core structures and exemplary modifications. (**A**) Structure of blumenol A, blumenol B and blumenol C. (**B**) Exemplary blumenol C derivatives. Glyc, glycoside.

DOI: https://doi.org/10.7554/eLife.37093.003

additionally malonylated or contain a 3-hydroxy-3-methylglutarate decoration (*Strack and Fester, 2006*; *Schliemann et al., 2008*). The connections among sugar components can also vary (e.g., glucose-(1''→4')-glucose or glucose-(1''→6')-glucose; *Maier et al., 2000*; *Fester et al., 2002*). The particular type of decorations appears to be highly species-specific and it is likely that additional structural variants remain to be discovered. Exemplary structures are shown in *Figure 1B*.

Interestingly, blumenols such as blumenol A, blumenol A-9-*O*-Glc, blumenol B, blumenol C and blumenol C-9-*O*-Glc, were also reported to occur in the aerial parts of various plant species (*Galbraith and Horn, 1972*; *Bhakuni et al., 1974*; *Takeda et al., 1997*). However, most of these studies focused on the identification of natural products using large scale extractions (up to several kg of plant material) and were not performed in the context of AMF colonization. Furthermore, some blumenol compounds were also found in plant families that are known to have lost their ability to establish AMF interactions (Brassicaceae: *Cutillo et al., 2005*; Urticaceae: *Aishan et al., 2010*). These reports indicate AMF-independent constitutive levels of particular blumenols in aerial plant parts. *Adolfsson et al., 2017* analyzed blumenol accumulations together with other metabolites in leaves of plants with and without AMF colonization. None of these studies reported AMF-specific accumulations of blumenols or transcripts specific for their biosynthesis. The concentrations of some blumenol derivatives were even reported to be down-regulated in response to AMF colonization (*Adolfsson et al., 2017*).

The identification of a reliable metabolite marker in aerial plant tissues would be highly useful for AMF research since the characterization of AMF-associations is still laborious and time-consuming, typically requiring destructive root harvesting and microscopic examination or transcript analyses (*Vierheilig et al., 2005*; *Parádi et al., 2010*). To identify readily accessible AMF-indicative shoot metabolites, we hypothesized that a subset of the AMF-induced root metabolites would accumulate in shoots as a result of transport or systemic signaling.

## Results

### Blumenols are AMF-indicative metabolic fingerprints in roots

We performed an untargeted metabolomics analysis of root tissues in a transgenic line of *Nicotiana attenuata*, silenced in the calcium- and calmodulin-dependent protein kinase (ir*CCaMK*) and empty vector (EV) plants co-cultured with or without *Rhizophagus irregularis* (*Figure 2A*). By using ir*CCaMK* plants, unable to establish a functional AMF-association (*Groten et al., 2015*), we were able to dissect the AMF association-specific metabolic responses from those changes that result from more general plant-fungus interactions. Untargeted metabolome profiling of roots using liquid chromatography (LC) coupled to time-of-flight mass spectrometry (qTOF-MS) resulted in a concatenated

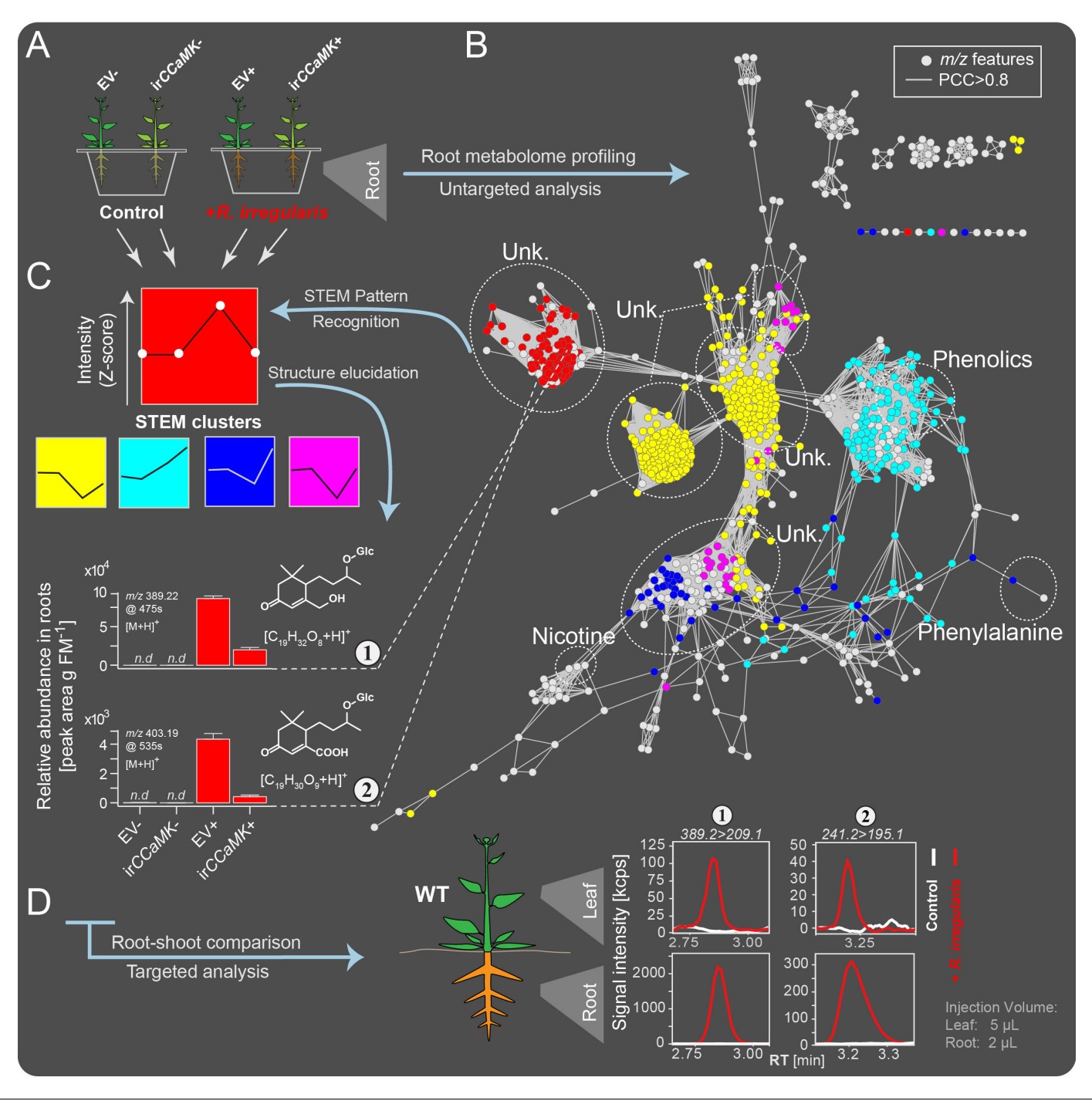

**Figure 2.** Combined targeted and untargeted metabolomics identified blumenol derivatives as AMF-indicative *in-planta* metabolic fingerprints in the roots and leaves of *Nicotiana attenuata* plants. (A) Experimental set-up. EV and ir*CCaMK* plants were co-cultured and inoculated with or without *R. irregularis*. Six weeks after inoculation (wpi), root samples were harvested for metabolite profiling. (B) Covariance network visualizing *m/z* features from UHPLC-qTOF-MS untargeted analysis (n = 8). Known compounds, including nicotine, phenylalanine and various phenolics, and unknown metabolites (Unk.) are enclosed in dashed ellipses. (C) Normalized Z-scores of *m/z* features were clustered using STEM Clustering; 5 of 8 significant clusters are shown in different colors and mapped onto the covariance network. The intensity variation (mean +SE) of 2 selected features (Compounds 1 and 2) are shown in bar plots (n.d., not detected). (D) Representative chromatograms of Compounds 1 and 2 in roots and leaves of plants with and without AMF inoculation, as analyzed by targeted UHPLC-triple quadrupole-MS metabolomics.

DOI: https://doi.org/10.7554/eLife.37093.004

The following source data and figure supplements are available for figure 2:

*Figure 2 continued*

**Source data 1.** Source data for *Figure 2*.
DOI: https://doi.org/10.7554/eLife.37093.007
**Figure supplement 1.** Abundance of root blumenol derivatives correlates positively with root AMF colonization.
DOI: https://doi.org/10.7554/eLife.37093.005
**Figure supplement 1—source data 1.** Source data for Source data for *Figure 2—figure supplement 1*.
DOI: https://doi.org/10.7554/eLife.37093.006

data matrix consisting of 943 mass features (*m/z* signals detected at particular retention times). A co-expression network analysis was conducted in which nodes represent *m/z* features and edges connect metabolite mass features originating from similar in-source fragmentations and sharing biochemical relationships (*Li et al., 2015*; *Li et al., 2016*). For example, features representing well-known compounds, like nicotine and phenylalanine, were tightly connected (*Figure 2B*). A STEM clustering pipeline was performed to recognize patterns of metabolite accumulations in the genotype × treatment data matrix [(EV/ir*CCaMK*) × (-/+AMF inoculation)]. As a result, 5 of 8 computed distinct expression patterns were mapped onto the covariance network in *Figure 2B* (shown in different colors). A tightly grouped cluster of unknown metabolites, highlighted in red (*Figure 2B*) occupied a distinct metabolic space. Metabolites grouped in this cluster were highly elicited upon mycorrhizal colonization in EV, but not in ir*CCaMK* plants and not found in plants without AMF associations (*Figure 2C*). The structures of the compounds of this cluster were annotated based on tandem-MS and NMR data. Five metabolites were annotated as blumenols: 11-hydroxyblumenol C-9-*O*-Glc (*Figure 2C*; Compound 1), 11-carboxyblumenol C-9-*O*-Glc (*Figure 2C*; Compound 2), 11-hydroxyblumenol C-9-*O*-Glc-Glc (Compound 3), blumenol C-9-*O*-Glc-Glc (Compound 4) and blumenol C-9-*O*-Glc (Compound 5).

To quantify these compounds throughout the plant, we used a more sensitive and specifically targeted metabolomics approach based on LC-triple-quadrupole-MS. The abundance of the five blumenol C-glycosides continually increased with mycorrhizal development (*Figure 2—figure supplement 1A*) and was highly correlated with the mycorrhizal colonization rate as determined by the transcript abundances of classical arbuscular mycorrhizal symbiosis-marker genes (fungal housekeeping gene, *Ri-tub; plant* marker genes, *Vapyrin*, *RAM1*, *STR1* and *PT4*; *Park et al., 2015*; *Figure 2—figure supplement 1B*, Data Set 1).

## Hydroxy- and carboxyblumenol C-glucoside levels in leaves positively correlate with root colonizations

Compounds 1 and 2 showed a similar AMF-specific accumulation in the leaves, as observed in the roots (*Figure 2D*). The other analyzed blumenols were not detected in leaves (Compounds 3 and 4; *Figure 3—figure supplement 1A*) or showed a less consistent AMF-specific accumulation (Compound 5; due to its constitutive background level; *Figure 3—figure supplement 1A*). The identity of Compounds 1 and 2 in the leaves was verified by high resolution qTOF-MS (*Figure 3—figure supplement 1B–E*).

Next, we determined the correlations between the contents of AMF-indicative foliar Compounds 1 and 2 and root colonization rates. In a kinetic experiment, the amount of both compounds steadily increased in the leaves of plants inoculated with *R. irregularis* (*Figure 3A*, *Figure 3—figure supplement 2*). At three wpi, the abundance of compounds 1 and 2 in the leaves was sufficient to reflect the colonization level of the roots. In contrast, the classical AMF-marker-genes, which are usually analyzed in the roots, did not respond in the leaves (*Figure 4*). In an inoculum-gradient experiment using increasing inoculum concentrations, proportionally higher Compound 1 and 2 levels were observed (*Figure 3B*), accurately reflecting the differential colonization of roots across treatments (*Figure 3E*). In addition to inoculation with a single AMF species (*R. irregularis*), we also tested mycorrhizal inoculum originally collected from the plant's native habitat, the Great Basin Desert in Utah, USA, which mainly consists of *Funneliformis mosseae* and *R. irregularis*. EV plants inoculated with this 'natural inoculum' also accumulated Compounds 1 and 2 in leaves, while ir*CCaMK* plants did not (*Figure 3C*). The analysis of a second independently transformed irCCaMK line confirmed the result that when the association with *R. irregularis* was genetically abrogated, Compounds 1 and 2 failed to accumulate in leaves of plants co-cultured with the AMF (*Figure 3—figure supplement*

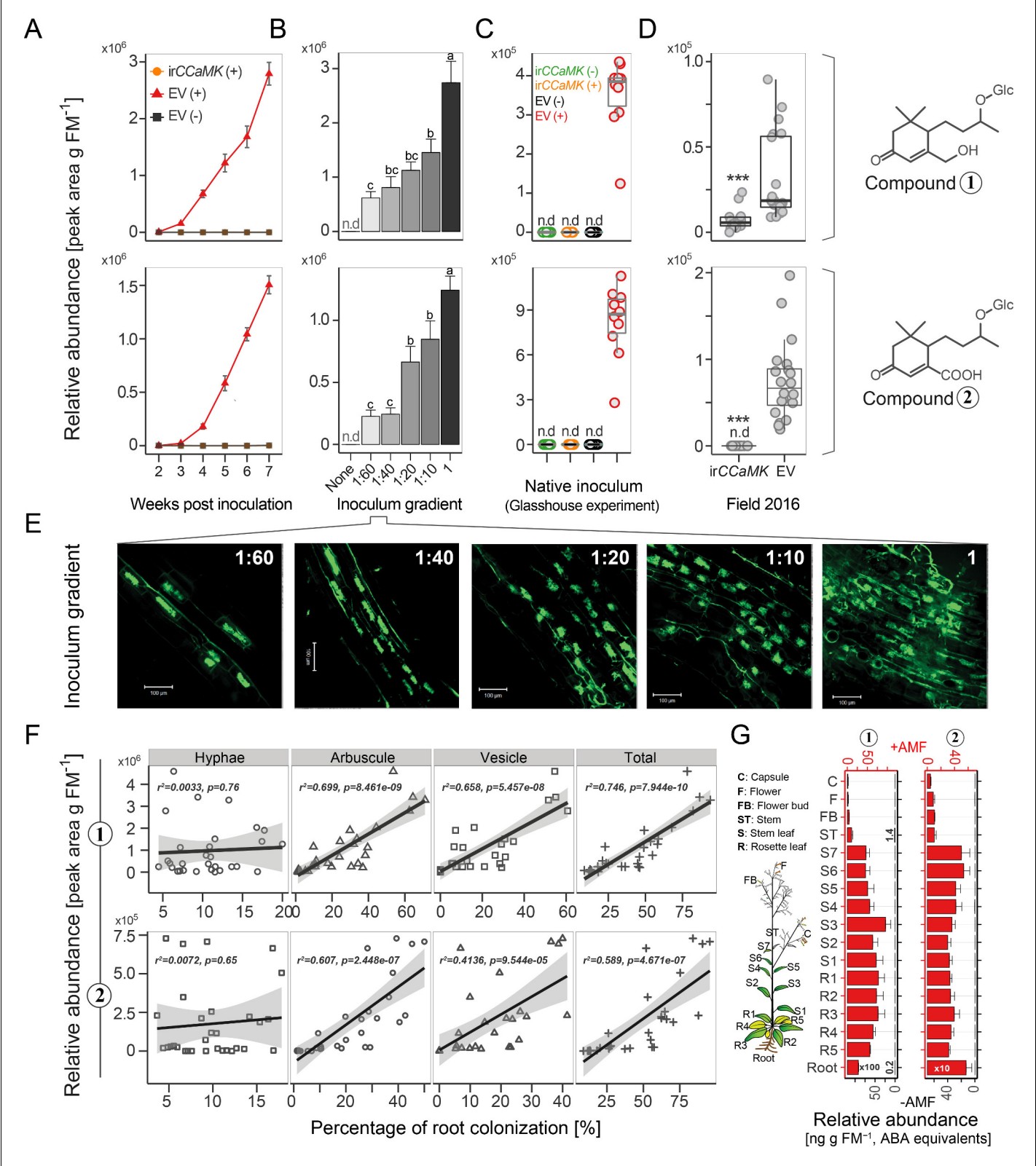

**Figure 3.** Compounds 1 and 2 are leaf markers of root AMF colonization in *N. attenuata*. (**A**) Time lapse accumulations of Compounds 1 and 2 in leaves of EV plants with (EV+, red) or without (EV-, black) AMF-inoculation and of ir*CCaMK* plants with AMF-inoculation (ir*CCaMK*+, orange, covered by black) (means ± SE, n ≥ 5). (**B**) Leaf abundances of Compounds 1 and 2 (five wpi) of plants inoculated with different inoculum concentrations (means + SE, n ≥ 4); different letters indicate significant differences ($p<0.05$, one-way ANOVA followed by Fisher's LSD). (**C**) Compounds 1 and 2 in leaf samples of

*Figure 3 continued*

EV and ir*CCaMK* plants inoculated with (+) or without (-) AMF inoculum isolated from the plant's native habitat (six wpi); different letters indicate significant differences (p<0.05, one-way ANOVA followed by Tukey's HSD, n = 10). (D) Field experiment (Great Basin Desert, Utah, USA): Compounds 1 and 2 in leaf samples of EV (n = 20) and ir*CCaMK* (n = 19) plants sampled eight weeks after planting. (Student's *t*-test: ***p<0.001). (E) Representative images of WGA-488 stained roots of plants shown in B) (bar = 100 µm). (F) Leaf Compounds 1 and 2 relative to the percentage of root colonization by hyphae, arbuscules, vesicles and total root length colonization of the same plants (linear regression model). (G) Compounds 1 and 2 in 17 different tissues of plants with (+AMF, n = 3, red bars) or without (-AMF, n = 1, black bars) AMF-inoculation harvested at six wpi.

DOI: https://doi.org/10.7554/eLife.37093.008

The following source data and figure supplements are available for figure 3:

**Source data 1.** Source data for *Figure 3*.
DOI: https://doi.org/10.7554/eLife.37093.015
**Figure supplement 1.** AMF-induced accumulation of blumenol derivatives in roots and leaves of *N. attenuata*.
DOI: https://doi.org/10.7554/eLife.37093.009
**Figure supplement 2.** Time course analysis of the root colonization by AMF and the corresponding accumulation of Compounds 1 and 2 in roots and leaves of *N. attenuata*.
DOI: https://doi.org/10.7554/eLife.37093.010
**Figure supplement 2—source data 1.** Source data for Source data for *Figure 3—figure supplement 2*.
DOI: https://doi.org/10.7554/eLife.37093.011
**Figure supplement 3.** Root AMF colonization and abundance of Compound 1 in a second independently transformed ir*CCaMK* line.
DOI: https://doi.org/10.7554/eLife.37093.012
**Figure supplement 3—source data 1.** Source data for Source data for *Figure 3—figure supplement 3*.
DOI: https://doi.org/10.7554/eLife.37093.013
**Figure supplement 4.** Signals from Compound 1 are partially disturbed in field samples, but not for Compound 2.
DOI: https://doi.org/10.7554/eLife.37093.014

*3*). When planted into the plant's natural environment in Utah, both EV and ir*CCaMK* plants could be clearly distinguished by their leaf Compound 1 and 2 contents. The signature of Compound 2 provided a better quality marker in these field-grown plants (*Figure 3D*, *Figure 3—figure supplement 4*). The foliar contents of these two compounds were highly correlated with the percentage of

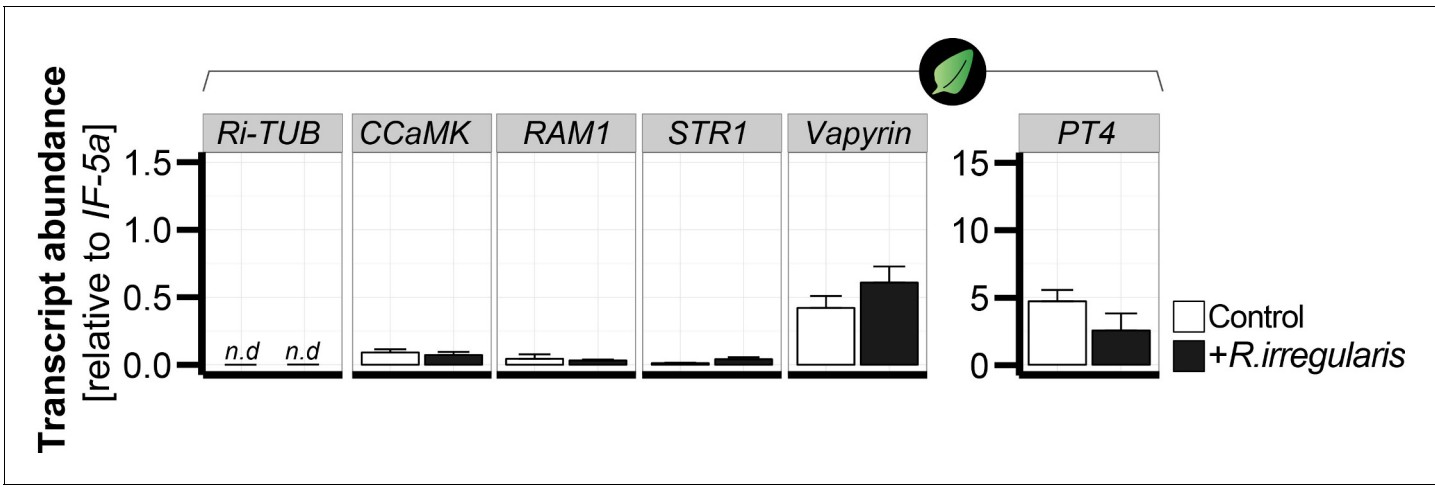

**Figure 4.** Transcript abundance of classical arbuscular mycorrhizal symbiosis-marker genes do not respond in leaves of mycorrhizal and control *N. attenuata* plants. The transcript abundance (relative to *NaIF-5a*) of classical root marker genes was analyzed in leaves of *N. attenuata* plants in the presence (+*R. irregularis*, black bars) and absence (control, white bars) of root colonization with *R. irregularis*. The marker genes include the *R. irregularis* specific housekeeping gene, *Ri-tub*, as well as the plant-derived marker genes *CCaMK*, *Vapyrin*, *PT4*, *STR1* and *RAM1*. Leaf samples were harvested six wpi and analyzed by qPCR. Data represent means +SE (n ≥ 3), n.d., not detected.

DOI: https://doi.org/10.7554/eLife.37093.016

The following source data is available for figure 4:

**Source data 1.** Source data for *Figure 4*.
DOI: https://doi.org/10.7554/eLife.37093.017

arbuscules in roots, the core structure of AMF interactions (*Figure 3F*, *Figure 3—figure supplement 2*). In contrast, other biotic or abiotic stresses, including herbivory, pathogen infection and drought stress, did not elicit the foliar accumulations of Compounds 1 and 2 (*Figure 5*). Such stimuli also do not induce blumenol accumulation in roots (*Maier et al., 1997*). An analysis of various plant tissues, including different leaf positions, stem pieces, flowers and capsules revealed that these AMF-specific signatures accumulated throughout the shoot (*Figure 3G*). Taken together, we conclude that the contents of particular blumenols in aerial plant parts robustly reflect the degree of mycorrhizal colonization in *N. attenuata* plants.

## AMF–indicative blumenols in shoots most likely originate from the roots

Blumenols are apocarotenoids originating from a side branch of the carotenoid pathway (*Hou et al., 2016*). Most of the candidate genes for blumenol biosynthesis were upregulated in roots, but not in leaves of *N. attenuata* plants in response to mycorrhizal colonization (*Figure 6A*, *Figure 6—figure supplement 1A*). We inferred that these AMF-indicative leaf apocarotenoids are transported from their site of synthesis in colonized roots to other plant parts. This is consistent with the occurrence of blumenols in stem sap (*Figure 6—figure supplement 1B*) which was collected by centrifuging small stem pieces. To clarify the origins (local biosynthesis vs. transport) of these leaf blumenols, we genetically manipulated the carotenoid biosynthesis of *N. attenuata* plants. To minimize the effects of a disturbed carotenoid biosynthesis on the AMF-plant interaction, we used the dexamethasone (DEX)-inducible pOp6/LhGR system to silence phytoene desaturase (PDS) expression in a single DEX-

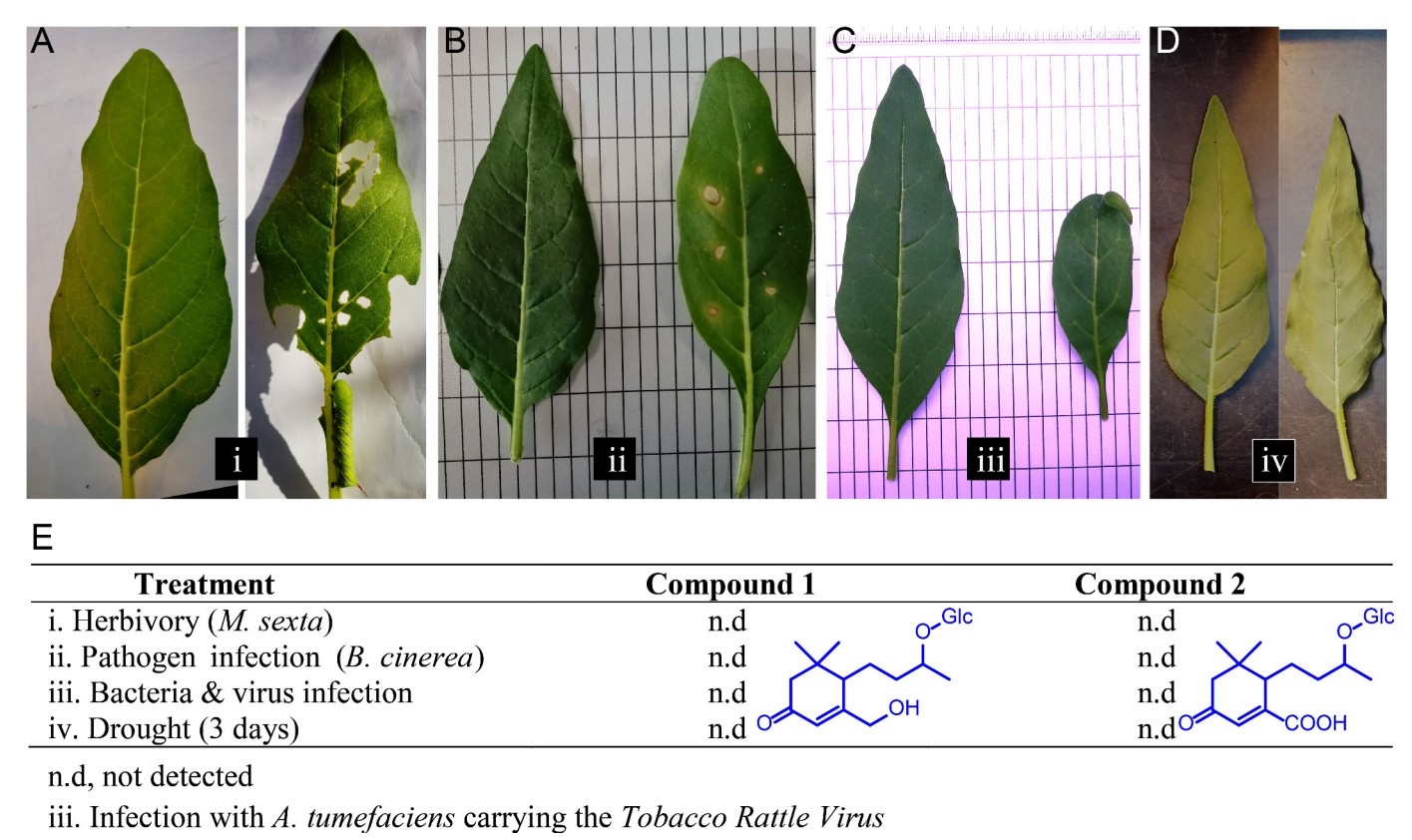

**Figure 5.** Different biotic and abiotic stresses do not elicit accumulations of Compounds 1 and 2 in leaves. (A–D) Representative leaves of *N. attenuata* plants subjected to different stresses (right leaf), as well as the untreated controls (left leaf): (A) *Manduca sexta* feeding for 10 days; (B) *Botrytis cinerea* infection for five days. (C) Infection for two weeks with *Agrobacterium tumefaciens* carrying the Tobacco Rattle Virus; (D) Dehydration for three days. For each treatment, four biological replicates were used. (E) Accumulation of Compounds 1 and 2 in treated samples from (A–D). n.d., not detected.
DOI: https://doi.org/10.7554/eLife.37093.018

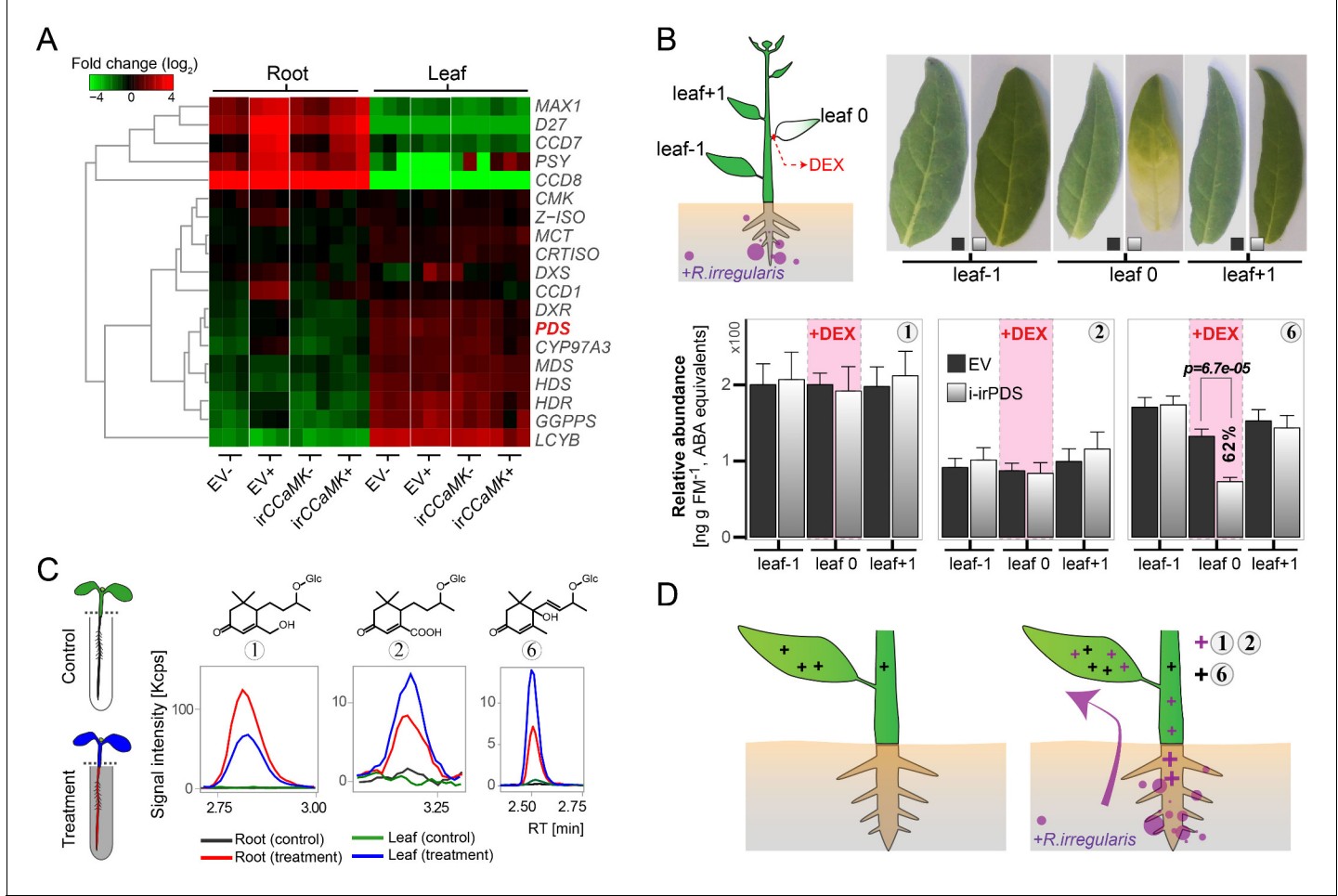

**Figure 6.** AMF-indicative Compounds 1 and 2 in shoots of mycorrhizal plants originate from the roots. (**A**) Hierarchical clustering analysis of transcript abundance from RNA-seq of methylerythritol 4-phosphate (MEP) and (apo)carotenoid biosynthetic genes (for details see *Figure 6—figure supplement 1A*). (**B**) Compounds 1, 2 (AMF-specific) and 6 (not AMF-specific) in AMF-inoculated i-irPDS and EV plants. On each plant, a single stem leaf (leaf 0) was elicited with 100 µM DEX-containing paste for three weeks; treated and adjacent, untreated control leaves (leaf −1 and leaf +1) were harvested. Representative leaves are shown (bleaching indicates PDS silencing); (means +SE, n ≥ 6). The same leaf positions in i-irPDS and EV plants were compared by Student's *t*-tests. (**C**) Contents of Compounds 1, 2 and 6 in the roots and shoots of seedlings whose roots were dipped for 1 d into an aqueous solution with (treatment) or without (control) AMF-indicative blumenols. (**D**) Model of blumenol distribution in plants with (right panel) and without (left panel) AMF colonization. The model illustrates constitutive blumenols (e.g., Compound 6 in *N. attenuata*) and AMF-indicative ones (e.g., Compounds 1 and 2 in *N. attenuata*) and their inferred transport.

DOI: https://doi.org/10.7554/eLife.37093.019

The following source data and figure supplements are available for figure 6:

**Source data 1.** Source data for *Figure 6*.
DOI: https://doi.org/10.7554/eLife.37093.024

**Figure supplement 1.** Foliar levels of Compounds 1 and 2 are derived from roots.
DOI: https://doi.org/10.7554/eLife.37093.020

**Figure supplement 1—source data 1.** Source data for *Figure 6—figure supplement 1*.
DOI: https://doi.org/10.7554/eLife.37093.021

**Figure supplement 2.** Compound 6 is constitutively produced in shoots of *N. attenuata* and not indicative of AMF associations.
DOI: https://doi.org/10.7554/eLife.37093.022

**Figure supplement 2—source data 1.** Source data for *Figure 6—figure supplement 2*.
DOI: https://doi.org/10.7554/eLife.37093.023

treated leaf position (*Schäfer et al., 2013*). Treated leaves showed clear signs of bleaching, indicating PDS silencing (*Figure 6B*, *Figure 6—figure supplement 1C*), but levels of the AMF-indicative Compounds 1 and 2 were not affected, consistent with their transport from other tissues, likely the highly accumulating roots. As a control, we analyzed the non-AMF-inducible Compound 6, showing constitutive levels in aerial tissues (*Figure 6—figure supplement 2*). In DEX-treated leaves, Compound 6 concentrations were reduced by nearly 40%, consistent with local production (*Figure 6B*, *Figure 6—figure supplement 1D*). To confirm the within-plant transport potential of blumenols, we dipped roots of seedlings into aqueous solutions of Compounds 1 or 2. After overnight incubation, the blumenol derivatives were clearly detected not only in roots, but also in shoots (*Figure 6C*, *Figure 6—figure supplement 1E*).

## The analysis of AMF-indicative blumenols as HTP screening tool for forward genetics approaches

To test the potential of the foliar AMF-indicative metabolites as a screening tool, we quantified the concentration of Compounds 1 and 2 in leaves of plants of a population of recombinant inbred lines (RILs) of a forward genetics experiment, an experiment which would be challenging with the classical screening tools of root staining or nucleic acid analysis. We focused our analysis on Compound 2 due to the superior quality of its signature in the leaves of field-grown plants (*Figure 3—figure supplement 4*). The experiment consisted of a population of RILs from a cross of two *N. attenuata* accessions (Utah, UT and Arizona, AZ) (*Zhou et al., 2017*) which differ in their mycorrhizal colonization (*Figure 7A–B*, *Figure 7—figure supplement 1*) and accumulation of foliar Compound 2 in the glasshouse (*Figure 7C*). A QTL analysis of 728 plants grown across a 7200 m$^2$ field plot (*Figure 7D*) revealed that the abundance of Compound 2 mapped to a single locus on linkage group 3 (*Figure 7E*), which harbored a homologue of *NOPE1* (*NIATv7_g02911*) previously shown to be required for the initiation of AMF symbioses in maize and rice (*Nadal et al., 2017*). Transcripts of *NaNOPE1* were more abundant in AZ roots after AMF inoculation, but did not differ significantly in leaves (*Figure 7—figure supplement 1B*). While clearly requiring additional follow-up work, these results highlight the value of these signature metabolites for HTP screens, which form the basis of most crop improvement programs.

## AMF-indicative blumenols in shoots are a widespread response of various plant species to different kinds of AMF

The AMF-specific accumulation of blumenol C-derivatives in roots is a widespread phenomenon within higher plants (*Strack and Fester, 2006*); however, how general are the observed blumenol changes in aerial parts across different combinations of plants and AMF species? We analyzed *Solanum lycopersicum*, *Triticum aestivum* and *Hordeum vulgare* plants with and without AMF inoculation and again we found an overlap in the AMF-specific blumenol responses in roots and leaves, consistent with the transport hypothesis. Further analyses led to the identification of additional AMF-indicative blumenols in the leaves of *Medicago truncatula*, *Solanum tuberosum* and *Brachypodium distachyon*. We identified various types of blumenols that showed an AMF-specific accumulation in the shoot, including blumenol B (Compound 7), which has not previously been reported in an AMF-dependent context (*Figure 7F*; *Figure 7—figure supplement 2*). As reported for roots, the particular blumenol types were species-dependent, but the general pattern was widespread across monocots and dicots in experiments conducted at different research facilities. In tests with diverse fungal species (*R. irregularis*, *F. mosseae* and *Glomus versiforme)*, the observed effects were not found to be restricted to specific AMF taxa (*Figure 7F*; *Figure 7—figure supplement 2*). In short, the method is robust.

## Discussion

AMF-interactions are proposed to have played an important role for the colonization of land by plants and still play an important role for a majority of plants by improving the function of their roots and increasing whole-plant performance. Consequently, the investigation of AMF-mediated effects on a host plant's physiology has been an important research field for many decades and characteristic transcriptional and metabolic changes have been observed in the roots of AMF-colonized plants. However, the cumbersome analysis of AMF-interactions, involving destructive harvesting of root

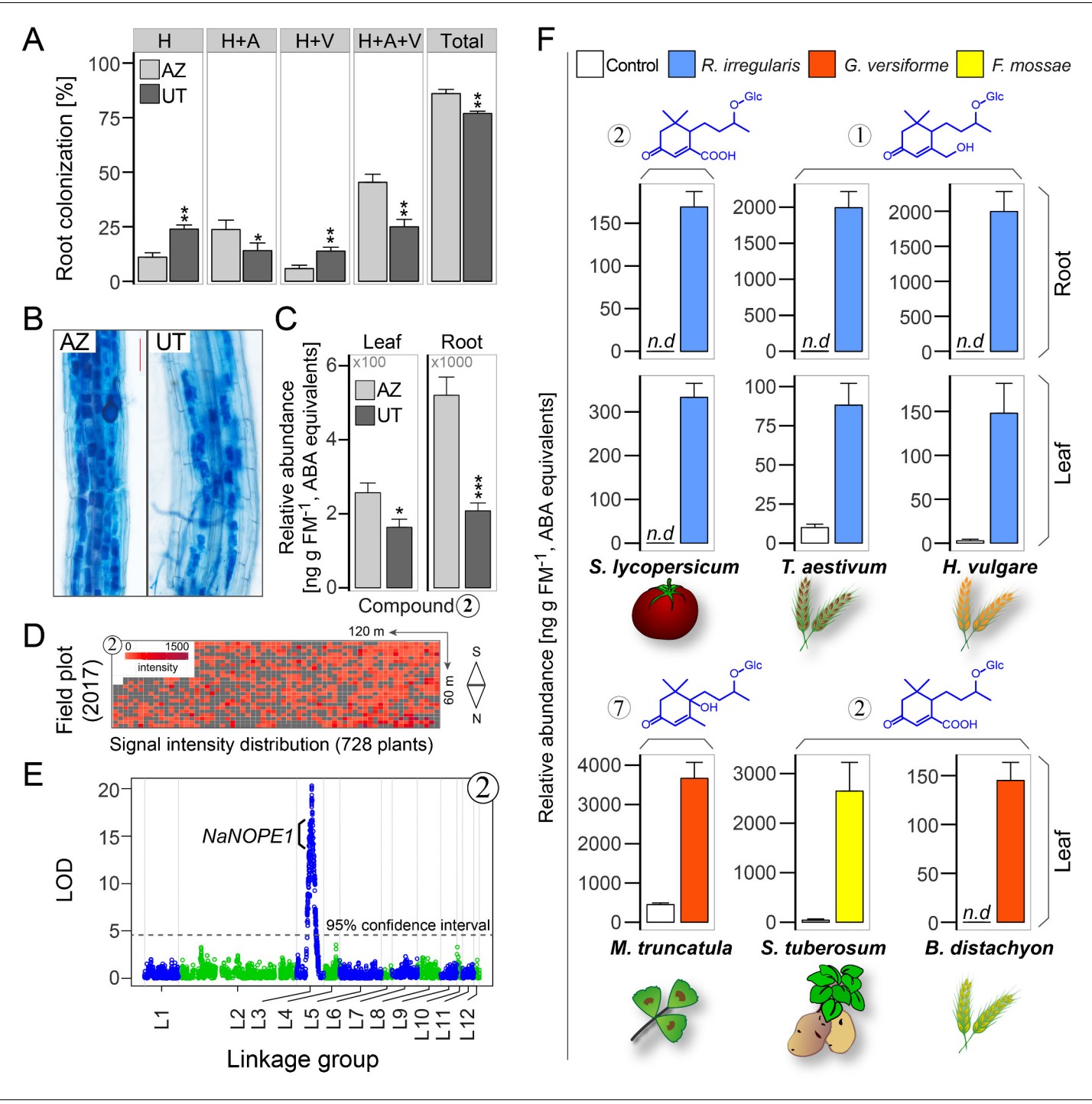

**Figure 7.** AMF-indicative changes in blumenols in aerial plant parts are valuable research tools providing accurate assessments of functional AMF associations in high-throughput screenings of multiple plant and AMF species. (**A**) Root colonization analysis in two *N. attenuata* accessions (UT/AZ). H: hyphae; A: arbuscules; V: vesicles; Total: total root length colonization (n = 4; Student's *t*-test, *p<0.05, **p<0.01, ***p<0.001). (**B**) Representative images of trypan blue stained roots (six wpi; bar = 100 μm). (**C**) Compound 2 in roots and leaves of UT and AZ plants with and without AMF-inoculation (means +SE, n = 8). (**D**) Heatmap of the normalized abundance of foliar Compound 2 in plants of a UT-AZ RIL population (728 plants) planted across a 7,200 m² field plot. (**E**) QTL mapping analysis of the data from D. The QTL on linkage group three contains Na*NOPE1*, an AMF-associated gene, in addition to others. LOD, logarithm of the odds ratio. (**F**) Blumenol contents of different crop and model plants with and without AMF inoculation (*S. lycopersicum* (n = 6), *T. aestivum* (n = 10), *H. vulgare* (n = 5): eight wpi; *M. truncatula* (n = 3): seven wpi; *S. tuberosum* (n = 5): six wpi; *B. distachyon* (n = 4): five wpi). Different plant and AMF species were used as indicated (means +SE; n.d., not detected).

DOI: https://doi.org/10.7554/eLife.37093.025

*Figure 7 continued*

The following source data and figure supplements are available for figure 7:

**Source data 1.** Source data for *Figure 7*.
DOI: https://doi.org/10.7554/eLife.37093.030
**Figure supplement 1.** Phenotypes of UT and AZ accessions and field plot planting design.
DOI: https://doi.org/10.7554/eLife.37093.026
**Figure supplement 1—source data 1.** Source data for *Figure 7—figure supplement 1*.
DOI: https://doi.org/10.7554/eLife.37093.027
**Figure supplement 2.** AMF-indicative changes in blumenols in aerial plant part are valuable research tools providing accurate assessments of functional AMF associations of multiple plant and AMF species (continued from *Figure 7F*).
DOI: https://doi.org/10.7554/eLife.37093.028
**Figure supplement 2—source data 1.** Source Code File for QTL analysis.
DOI: https://doi.org/10.7554/eLife.37093.029

tissues and microscopic or transcript analysis, restrains large-scale investigations and commercial applications. AMF interactions were also shown to affect the primary and secondary metabolism in the systemic, aerial tissues of plants; however none of these responses proved to be widespread and sufficiently specific to function as reliable markers (*Schweiger and Müller, 2015*). Here we describe the discovery of particular blumenols as AMF-indicative markers in leaves and other systemic aerial tissues and illustrate their potential application as tools for research and plant breeding.

## Systemic AMF-mediated metabolite changes

Metabolites and metabolite responses are often specific to particular parts and tissues of a plant (*Li et al., 2016*; *Lee et al., 2017*), but it is also known that local responses can spread to other plant parts. Additionally, metabolites do not only accumulate at their place of biosynthesis but can be readily transported throughout the plant (*Baldwin, 1989*). Therefore, we hypothesized that local changes in the roots might also be reflected in the systemic aerial tissues, either by signaling or transport. This allowed us to identify specific AMF-indicative blumenols in the shoot (*Figure 3*) despite the occurrence of other highly abundant and constitutively produced compounds and blumenols that are not indicative of AMF-associations. Interestingly, the confirmation of compound identities in leaf samples with high resolution MS techniques proved to be challenging and required additional sample purification steps. Likely, such matrix effects thwarted the detection of these AMF-indicative, systemic blumenol responses in previous investigations. Under our conditions, AMF-indicative blumenols began mirroring the colonization rates of roots at around three wpi (*Figure 3—figure supplement 2*). Although microscopic methods can detect first signs of AMF colonization of the roots already after a few days (*Brundrett et al., 1985*), the sensitivity of our method was found to be sufficient to analyze colonization rates observed in the glasshouse and, more importantly, in nature (*Figures 3* and *7*). The discovery of these AMF-indicative blumenol compounds in diverse plant species interacting with different AMF species (*Figure 7*, *Figure 7—figure supplement 2*) further indicates that these responses are widespread. AMF-induced blumenols in the roots have been shown to be quantifiable by various MS and photodiode array based detector setups. However, AMF-induced blumenols occur in many-fold lower amounts in the leaves compared to the roots (e. g., Compound 1 in *N. attenuata* approximately 1/10) and, at these low concentrations, their analysis is likely thwarted by complex matrix effects. Therefore, their analysis requires detection systems with advanced sensitivity and selectivity as is offered by state-of-the-art triple quadrupole technology and enhanced sample preparations (e.g., by solid-phase-extraction based purification and concentration) which were used in the quantitative detection of leaf blumenols described here. In addition to the blumenols, mycorradicin, another biosynthetically related type of apocarotenoids, was reported to accumulate in AMF-colonized roots (*Klingner et al., 1995*) and it would be interesting to investigate if mycorradicin accumulates throughout the plant, as well.

## Root-to-shoot transport of AMF-indicative blumenols

Despite the AMF-induced accumulation of blumenols in the shoot, putative candidate genes of the apocarotenoid biosynthesis pathway were only induced in the roots of AMF-inoculated plants

(*Figure 6A*, *Figure 6—figure supplement 1A*). To exclude other mechanisms (e.g., post-transcriptional regulation) mediating the local production of the blumenol compounds in the leaves, we genetically manipulated the carotenoid pathway in a tissue-specific manner. It is challenging to manipulate blumenols without affecting the AMF-colonization of the plant, since other carotenoid-derived compounds, such as strigolactones, are known to play an important role in this process (*Lanfranco et al., 2018*). To circumvent these problems, we used the LhGR/pOp6 system for chemically inducible RNAi-mediated gene silencing of PDS (*Schäfer et al., 2013*) to impair carotenoid biosynthesis only in a particular leaf of AMF-inoculated plants. Interestingly, only the constitutively produced Compound 6 was reduced in the treated leaves, while the AMF-indicative Compounds 1 and 2 were not affected by our treatment (*Figure 6B*). This indicated that instead of being locally produced, Compound 1 and 2 are translocated from the roots, an inference consistent with the occurrence of AMF-indicative blumenols in stem sap and the capacity of seedlings to transport blumenols from the root to the shoot from hydroponic solution (*Figure 6C*, *Figure 6—figure supplement 1B,D*). It seems likely that the AMF-indicative blumenols are transported in the xylem with the transpiration stream (*Figure 6D*). The blumenol glucosides (Compounds 1, 2 and 6) are hydrophilic low-molecular weight (402, 388 and 386 Da) compounds that are unlikely to pass membranes without further support, e.g., by transporters. It was recently demonstrated that ATP-binding cassette (ABC) transporters (G-type) are involved in the root-to-shoot transport of ABA, a phytohormone with a molecular structure related to blumenols, and these transporters resemble the function of other ABCG-type proteins reported to mediate the long-distance transport of cytokinins and strigolactones (*Borghi et al., 2015*). Whether blumenols are transported by similar mechanisms remains an interesting question.

## Functional implication of blumenol accumulation and transport

Blumenols were shown to accumulate in large amounts in the roots of various plants after AMF-inoculation (*Strack and Fester, 2006*) and our data indicate that they are subsequently distributed throughout the plant (*Figure 3G*). While the conservation of this response in various plants after inoculation with different AMF species (*Figure 7F*, *Figure 7—figure supplement 2*) indicates an important functional role in the AMF-plant interaction, this function remains to be explored. Unfortunately, the current knowledge of the biological activity of blumenols only vaguely indicates potential systemic functions of AMF-induced blumenols in shoot tissues. Activity studies on vomifoliol, the aglycone of the not AMF-indicative Compound 6, showed that this compound induces stomatal closure similar to the structurally related abscisic acid (*Stuart and Coke, 1975*). Additionally, blumenols are known to suppress seed germination and plant growth (*Kato-Noguchi et al., 2012*; *Kato-Noguchi et al., 2015*). Therefore, AMF-induced blumenols could serve as systemic signals that mediate the large-scale adjustments in general physiology that are thought to accompany AMF-interactions. For example, AMF-induced blumenols could be involved in the regulation of differential susceptibility of AMF-inoculated plants to stresses, such as drought or pathogen infection.

## AMF-indicative blumenols as tool for research and plant breeding

Even if classical tools for the quantification of AMF-plant interactions can offer superior sensitivity they are labor intensive and highly destructive which limits their application in studies that require high sample throughput, as well as in experiments that require repeated analysis of plants. We propose that the analysis of AMF-indicative blumenols in the shoot provides a convenient, easy-to-use, and minimally destructive tool to interrogate plant-AMF interactions in a HTP manner that allows for forward genetic studies even under field conditions (*Figure 7E*) and empowers plant breeding programs to produce mycorrhiza-responsive and P-efficient high-yielding lines (*van de Wiel et al., 2016*). Currently, phosphate fertilizer is derived from phosphate rock, a non-renewable resource, which is predicted to be depleted soon (*Vaccari and Strigul, 2011*). By enabling breeding programs to select crop varieties which have negotiated AMF symbioses that deliver high yields with minimal P inputs, this discovery could help steer the 'green revolution' away from intense agricultural inputs and the collateral environmental damage they cause. While some of the 'green revolution' crop varieties with gibberellin response defects are potentially more efficient in Pi uptake as a result of their higher root colonization rates by AMF (*Floss et al., 2013*; *Foo et al., 2013*) this serendipitous

breeding event underscores the value of explicitly designing crop breeding programs to produce crops that negotiate more favorable AMF associations.

## Materials and methods

### Key resources table

| Reagent type (species) or resource | Designation | Source or reference | Identifiers | Additional information |
|---|---|---|---|---|
| Genetic reagent (*N. attenuata*) | A-09-1212-1 | *Groten et al. (2015)*, DOI: 10.1111/pce.12561 | | Stably silenced in CCaMK via RNAi |
| Genetic reagent (*N. attenuata*) | A-09-1208-6 | *Groten et al. (2015)*, DOI: 10.1111/pce.12561 | | Stably silenced in CCaMK via RNAi |
| Genetic reagent (*N. attenuata*) | A-11-92−4 × A-11-325-4 | *Schäfer et al. (2013)*, DOI: 10.1111/tpj.12301 | | Chemically-inducible silenced in PDS via RNAi |
| Genetic reagent (*N. attenuata*) | A-04-266-3 | *Bubner et al. (2006)*, DOI: 10.1007/s00299-005-0111-4 | | Empty vector control |
| Biological sample (*N. attenuata*) | AZ-UT RIL | *Zhou et al. (2017)*, DOI: 10.1016/j.cub.2017.03.017 | | Biparental QTL mapping population |

### Plant material and AMF inoculation

For our experiments with *Nicotiana attenuata* (Torr. ex S. Wats.), we used plants from the 31st inbred generation of the inbred 'UT' line, ir*CCaMK* [A-09-1208-6 and A-09-1212-1(*Groten et al., 2015*) plants that are stably silenced in *CCaMK* via RNAi, i-ir*PDS* plants (A-11-92−4 × A-11-325-4; *Schäfer et al., 2013*) harboring the LhGR/pOp6 system for chemically-inducible RNAi-mediated gene silencing of phytoene desaturase (PDS) and the respective empty vector (EV) transformed plants (A-04-266-3; *Bubner et al., 2006*) as controls. Details about the transformation and screening of the ir*CCaMK* plants are described by *Groten et al. (2015)* and for the i-ir*PDS* plants by *Schäfer et al. (2013)*. Seeds were germinated on Gamborg B5 as described by *Krügel et al. (2002)*. The advance intercross recombinant inbred line (RIL) population was developed by crossing two *N. attenuata* inbred lines originating from accessions collected in Arizona (AZ) and Utah (UT), USA (*Glawe et al., 2003*; *Zhou et al., 2017*). Additionally, we used *Solanum lycopersicum* 'Moneymaker', *Hordeum vulgare* 'Elbany 'and *Triticum aestivum* 'Chinese Spring 'plants.

For glasshouse experiments, plants were treated according to *Groten et al. (2015)*. In brief, they were transferred into dead (autoclaved twice at 121°C for 30 min; non-inoculated controls) or living inoculum (*R. irregularis*, Biomyc Vital, inoculated plants) diluted 1:10 with expanded clay (size: 2–4 mm). Pots were covered with a thin layer of sand. Plants were watered with distilled water for 7 d and subsequently fertilized every second day either with a full strength hydroponic solution (for 1 L: 0.1292 g CaSO$_4$ × 2H$_2$O, 0.1232 g MgSO$_4$ × 7H$_2$O, 0.0479 g K$_2$HPO$_4$, 0.0306 g KH$_2$PO$_4$, 2 mL KNO$_3$ (1 M), 0.5 mL micronutrients, 0.5 mL Fe diethylene triamine pentaacetic acid) or with a low P hydroponics solution containing only 1/10 of the regular P-concentration (0.05 mM). Plants were grown separately in 1L pots, if not stated otherwise. In the paired design (*Figure 2*), ir*CCaMK* plants were grown together with EV plants in 2L pots and the watering regime was changed to ¼ of the regular P-concentration after plants started to elongate. Glasshouse experiments with natural inoculum (*Figure 3C*) were conducted in a mesocosm system (four boxes, each 2 pairs of EV and ir*CCaMK* plants). Plants were maintained under standard glasshouse conditions (16 hr light, 24–28°C, and 8 hr dark, 20–24°C and 45–55% humidity) with supplemental light supplied by high-pressure sodium lamps (Son-T-Agro).

The field experiments were conducted as described by *Schuman et al. (2012)*. Seedlings were transferred to Jiffy pots and planted into a field plot at the Lytle Ranch Preserve in the Great Basin Desert (Utah, USA: N 37.1412, W 114.0275). Field season 2016 (*Figure 3D*): field experiments were conducted under the US Department of Agriculture Animal and Plant Health Inspection Service (APHIS) import permission numbers 10-004-105m (ir*CCaMK*) and 07-341-101n (EV) and the APHIS release permission number 16-013-102r. EV and ir*CCaMK* plants were planted in communities of six plants, either of the same genotype or with both genotypes in equal number.

## Sample collection

During harvests, roots were washed and briefly dried with a paper towel. Subsequently, they were cut into 1 cm pieces and mixed. Plant tissues were shock-frozen in liquid nitrogen immediately after collection, ground to a fine powder and stored at −20°C (short-term storage)/−80°C (long-term storage) until extraction. From the root samples, an aliquot was stored in root storage solution (25% ethanol and 15% acetic acid in water) at 4°C for microscopic analysis.

For stem sap collection, branches of *N. attenuata* plants were cut into 1.5 cm long pieces and placed into small 0.5 mL reaction tubes with a small hole in the tip, which were placed in a larger 1.5 mL reaction tube. The tubes were centrifuged for 15 min at 10 000 × g. The stem sap from the larger reaction tubes was collected and stored at −20°C.

## Samples prepared at other laboratory facilities

*Medicago truncatula* (*Figure 7* and *Figure 7—figure supplement 2*) and *Brachypodium distachyon* (*Figure 7* and *Figure 7—figure supplement 2*) samples were prepared at the laboratory of Prof. Maria Harrison from the Boyce Thompson Institute for Plant Research (Ithaca, NY, USA). *M. truncatula* plants were grown in a growth chamber with a 16 hr light (25°C)/8 hr dark (22°C) cycle. *B. distachyon* plants were grown in growth chamber with a 12 hr light (24°C)/12 hr dark (22°C) cycle. All experiments were carried out in surface sterilized containers, autoclaved growth substrates and with surface sterilized spores and seeds as described previously (*Liu et al., 2004*; *Hong et al., 2012*; *Floss et al., 2013*). The growth substrates were mixtures of play sand (average particle 200–300 µm), black sand (heterogeneous particle size 50–300 µm) and gravel (heterogeneous particle size 300 µm −10 mm) as outlined below. For *M. truncatula*, 2 d-old seedlings were planted into 20.5 cm cones (Cone-tainers) containing a 1:1 mixture of sterile black sand and gravel with 200 surface-sterilized *G. versiforme* spores placed on a layer of play sand positioned 4 cm below the top of the cones. Five seedlings were planted into each cone. Plants were fertilized twice weekly with 20 mL of modified 1/2-strength Hoagland's solution (*Millner and Kitt, 1992*) containing 100 µM potassium phosphate. Plants were harvested 49 d post planting and tissue frozen in liquid nitrogen and stored at −80 C. One cone containing five seedlings represents one biological replicate. The harvest date was 3/3/2015. *B. distachyon* seedlings were planted into cones (Cone-tainers) containing a 2:1:1 mixture of black sand:play sand:gravel with 300 surface-sterilized *G. versiforme* spores placed on a layer of play sand positioned 4 cm below the top of the cones. Plants were fertilized twice weekly with 20 mL of modified 1/4-strength Hoagland's solution (*Millner and Kitt, 1992*) containing 20 µM potassium phosphate. Plants were harvested 35 d post planting and tissue frozen in liquid nitrogen and stored at −80 C. Each cone contained three plants and each biological replicate consisted of a pool of 4 cones. The harvest date was 6/20/2016.

*S. lycopersicum* 'Moneymaker '(*Figure 7—figure supplement 2*) and *Solanum tuberosum* 'Wega' (*Figure 7*) samples were prepared at the laboratory of Prof. Philipp Franken by Dr. Michael Bitterlich from the Leibniz-Institute of Vegetable and Ornamental Crops (Großbeeren/Erfurt Germany). *S. lycopersicum* were transplanted into 10 L open pots containing a sand/vermiculite mixture (sand: grain size 0.2–1 mm; Euroquarz, Ottendorf-Okrilla, Germany, vermiculite: agra vermiculite, Pullrhenen, Rhenen, The Netherlands; 1:1 v:v) and grown in the glass house from March to May (20-28:17 °C day:night, PAR: 300–2000 µmol m$^{-2}$ s$^{-1}$). Mycorrhizal plants were inoculated with a commercial inoculum either containing *R. irregularis* DAOM 197198 (INOQ, Schnega, Germany) or *F. mosseae* BEG12 (MycAgro Laboratory, Breteniere, France) with 10% of the substrate volume and were harvested after 11 or 6 weeks, respectively. *S. tuberosum* tubers of similar size were planted into 3 L pots filled with the same substrate and grown in a growth cabinet (20:16 °C day:night, 16 hr light, 8 hr dark; PAR: 250–400 µmol m$^{-2}$ s$^{-1}$, 50% rH). Mycorrhizal plants were inoculated with a commercial inoculum containing *F. mosseae* BEG12 (MycAgro Laboratory, Breteniere, France) with 10% of the substrate volume and were harvested after 6 weeks. Non-mycorrhizal counterparts were inoculated with the same amount of autoclaved (2 hr, 121°C) inoculum and a filtrate. The filtrate was produced for every pot by filtration of 200 mL deionized water through Whatman filter (particle retention 4–7 µm; GE Healthcare Europe GmbH, Freiburg, Germany) containing approx. 200 mL of inoculum. The same amount of deionized water (200 mL) was added to mycorrhizal pots. Plants were irrigated every other day with 400–600 mL nutrient solution (*De Kreij et al., 1997*); 40% of full strength) with 10% of the standard phosphate to guarantee good colonization (N: 10.32 mM; P: 0.07 mM, K: 5.5

mM, Mg: 1.2 mM, S: 1.65 mM, Ca: 2.75 mM, Fe: 0.02 mM, pH: 6.2, EC: 1.6 mS). For the experiment in the glasshouse, additional irrigation was carried out with deionized water until pot water capacity every other day. The pooled bulk leaf sample was dried at 60°C for 48 hr, ground to a fine powder and stored under dry conditions at room temperature until further analyses.

## Stress treatments

Herbivory treatments were conducted by placing *Manduca sexta* neonates, originating from an in-house colony, on the plants. After feeding for two weeks, rosette leaves were harvested. As controls, we harvested leaves from untreated plants.

For bacteria and virus infection, plants were inoculated with *Agrobacterium tumefaciens* carrying the Tobacco Rattle Virus. The inoculation was conducted by infiltrating leaves with a bacteria suspension using a syringe. The treatment was conducted as described for virus-induced gene silencing described by *Ratcliff et al. (2001)* and by *Saedler and Baldwin (2004)*. After incubation for three weeks, stem leaves of the treated plants and untreated control plants were harvested.

The fungal infection was done with *Botrytis cinerea*. On each plant, three leaves were treated by applying six droplets, each containing 10 µL of *B. cinerea* spore suspension ($10^6$ spores mL$^{-1}$ in Potato Extract Glucose Broth, Carl Roth GmbH), to the leaf surface. As control, plants were treated with broth without spores in the same way. Samples were collected after four days incubation.

Drought stress was induced by stopping the watering for four days. Subsequently, stem leaves of the drought-stressed plants and the continuously watered control plants were harvested. In contrast to the other samples of the stress experiment, leaves were dried before analysis to compensate for weight differences caused by changes in the water content.

## Sample preparation - extraction and purification

For extraction, samples were aliquoted into reaction tubes, containing two steel balls. Weights were recorded for later normalization. Per 100 mg plant tissues (10 mg in case of dry material), approximately 1 mL 80% MeOH was added to the samples before being shaken in a GenoGrinder 2000 (SPEX SamplePrep) for 60 s at 1150 strokes min$^{-1}$. After centrifugation, the supernatant was collected and analyzed. For triple-quadrupole MS quantification, the extraction buffer was spiked with stable isotope-labeled abscisic acid (D$_6$-ABA, HPC Standards GmbH) as an internal standard.

Stem sap was diluted 1:1 with MeOH spiked with D$_6$-ABA as an internal standard. After centrifugation, the supernatant was collected and analyzed.

The purification of *N. attenuata* leaf extracts for high resolution qTOF-MS was conducted by solid-phase-extraction (SPE) using Chromabond HR-XC 45 µm benzensulfonic acid cation exchange columns (Machery-Nagel) to remove abundant constituents, such as nicotine and phenolamides. After purification the samples were evaporated to dryness and reconstituted in 80% methanol.

Compound identification was conducted by NMR with purified fractions of root and leaf extracts. Compounds 1, 3 and 4 were extracted from root tissues of *N. attenuata* and purified by HPLC (Agilent-HPLC 1100 series; Grom-Sil 120 ODS-4 HE, C18, 250 × 8 mm, 5 µm; equipped with a Gilson 206 Abimed fraction collector). Compounds 2 and 7 were extracted from a mixture of leaf tissues from different plant species (*M. truncatula, Z. mays, S. lycopersicum* and *N. attenuata*). The first purification step was conducted by SPE using the Chromabond HR-XC 45 µm benzensulfonic acid cation exchange columns (Machery-Nagel) to remove hydrophilic and cationic constituents. Additional purification steps were conducted via HPLC (Agilent-HPLC 1100 series; Phenomenex Luna C18(2), 250 × 10 mm, 5 µm; equipped with a Foxy Jr. sample collector) and UHPLC (Dionex UltiMate 3000; Thermo Acclaim RSLC 120 C18, 150 × 2.1 mm, 2.2 µm; using the auto-sampler for fraction collection).

## Untargeted MS based analyses

For high resolution mass spectrometry (MS), indiscriminant tandem mass spectrometry (idMS/MS), tandem MS (MS$^2$) and pseudo-MS$^3$ were used. Ultra-high performance liquid chromatography (UHPLC) was performed using a Dionex UltiMate 3000 rapid separation LC system (Thermo Fisher), combined with a Thermo Fisher Acclaim RSLC 120 C18, 150 × 2.1 mm, 2.2 µm column. The solvent composition changed from a high % A (water with 0.1% acetonitrile and 0.05% formic acid) in a linear gradient to a high % B (acetonitrile with 0.05% formic acid) followed by column equilibration

steps and a return to starting conditions. The flow rate was 0.3 mL min$^{-1}$. MS detection was performed using a micrOTOF-Q II MS system (Bruker Daltonics), equipped with an electrospray ionization (ESI) source operating in positive ion mode. ESI conditions for the micrOTOF-Q II system were end plate offset 500 V, capillary voltage 4500 V, capillary exit 130 V, dry temperature 180°C and a dry gas flow of 10 L min$^{-1}$. Mass calibration was performed using sodium formiate (250 mL isopropanol, 1 mL formic acid, 5 mL 1 M NaOH in 500 mL water). Data files were calibrated using the Bruker high-precision calibration algorithm. Instrument control, data acquisition and reprocessing were performed using HyStar 3.1 (Bruker Daltonics). idMS/MS was conducted in order to gain structural information on the overall detectable metabolic profile. For this, samples were first analyzed by UHPLC-ESI/qTOF-MS using the single MS mode (producing low levels of fragmentation that resulted from in-source fragmentation) by scanning from m/z 50 to 1400 at a rate of 5000 scans s$^{-1}$. MS/MS analyses were conducted using nitrogen as collision gas and involved independent measurements at the following four different collision-induced dissociation (CID) voltages: 20, 30, 40 and 50 eV. The quadrupole was operated throughout the measurement with the largest mass isolation window, from m/z 50 to 1400. Mass fragments were scanned between m/z 50 to 1400 at a rate of 5000 scans s$^{-1}$. For the idMS/MS assembly, we used a previously designed precursor-to-product assignment pipeline (*Li et al., 2015*; *Li et al., 2016*) using the output results for processing with the R packages XCMS and CAMERA (Data Set 2).

Additional MS/MS experiments were performed on the molecular ion at various CID voltages. For the fragmentation of the proposed aglycones via pseudo-MS[3], we applied a 60 eV in-source-CID transfer energy which produced spectra reflecting the loss of all sugar moieties.

## Structure elucidation by NMR

Purified fractions were completely dried with $N_2$ gas and reconstituted with MeOH-$d_3$ prior to analysis by nuclear magnetic resonance spectroscopy (NMR). Structure elucidation was accomplished on an Avance III AV700 HD NMR spectrometer (Bruker-Biospin, Karlsruhe, Germany) at 298 K using a 1.7 mm TCI CryoProbe$^{TM}$ with standard pulse programs as implemented in Bruker TopSpin (Version 3.2). Chemical shift values (δ) are given relative to the residual solvent peaks at $δ_H$ 3.31 and $δ_C$ 49.05, respectively. Carbon shifts were determined indirectly from $^1$H-$^{13}$C HSQC and $^1$H-$^{13}$C HMBC spectra. The data are shown in *Table 1* and compared with previously published reference data (*Matsunami et al., 2010*). Blumenol C glucoside and byzantionoside B differ only in the configuration of position C-9; blumenol C glucoside is (9S)-configured whereas byzantionoside B has a (9R)-configuration. Characteristic $^{13}$C-chemical shift differences can thus be found for the positions C-9, C-10 and C-1'. In byzantionoside, C-9 and C-10 were reported to have chemical shifts of $δ_C$ 75.7 and $δ_C$ 19.9, respectively. In contrast, the chemical shifts for the same positions in blumenol C glucoside were reported to be lowfield shifted to $δ_C$ 77.7 and $δ_C$ 22.0, respectively. Experimental chemical shifts of C-9 for the compounds identified in this publication were in the range from $δ_C$ 77.2 to $δ_C$ 78.2, and for C-10 in the range from $δ_C$ 21.6 to $δ_C$ 21.9, respectively. C-1' of byzantionoside was reported to have a chemical shift of $δ_C$ 102.3, while for blumenol C glucoside the chemical shift was $δ_C$ 104.1. The experimental chemical shifts for C-1' of the compounds of this publication are in the range from $δ_C$ 103.8 to $δ_C$ 104.1. Hence the $^{13}$C-chemical shift data are completely consistent with the structures being blumenol C glucosides rather than byzantionoside B. More characteristic differences can be found in the $^1$H chemical shifts. The methylene shifts for H-7 of byzantionoside were reported to have chemical shifts of $δ_H$ 1.50 and $δ_H$ 1.98 while for blumenol C glucoside the same position showed chemical shifts of $δ_H$ 1.67 and $δ_H$ 1.81. Experimental $^1$H chemical shifts for H-7 of the compounds 1–4 of this publication were found in the range of $δ_H$ 1.62 to $δ_H$ 1.69 and $δ_H$ 1.80 to $δ_H$ 1.88, respectively. Consequently, the NMR data clearly establish the structures to be blumenol C derivatives and not byzantionosides.

## Targeted metabolite analysis

For chromatographic separations, a UHPLC (Dionex UltiMate 3000) was used to provide a maximum of separation with short run times. This reduced the interference from other extract components (matrix effects), increased the specificity of the method, and met the requirements of a HTP analysis. The auto-sampler was cooled to 10°C. As a stationary phase, we used a reversed phase column (Agilent ZORBAX Eclipse XDB C18, 50 × 3.0 mm, 1.8 μm) suitable for the separation of moderately polar

**Table 1.** $^1$H and $^{13}$C NMR data for compounds 1–4 and 7.

| No. | Compound 1 | | | Compound 2 | | | Compound 3 | | | Compound 4 | | | Compound 7 | | |
|---|---|---|---|---|---|---|---|---|---|---|---|---|---|---|---|
| Pos. | $\delta_H$ | mult., J [Hz] | $\delta_C$ | $\delta_H$ | mult., J [Hz] | $\delta_C$ | $\delta_H$ | mult., J [Hz] | $\delta_C$ | $\delta_H$ | mult., J [Hz] | $\delta_C$ | $\delta_H$ | mult., J [Hz] | $\delta_C$ |
| 1 | - | - | 202.3 | - | - | 203.1 | - | - | 202.2 | - | - | 202.3 | - | - | 200.9 |
| 2 | 6.06 | dd, 1.8/1.8 | 121.3 | 6.40 | s | 128.5 | 6.06 | s br | 121.3 | 5.81 | s br | 125.2 | 5.82 | s | 126.4 |
| 3 | - | - | 172.4 | - | - | 172.6 | - | - | 172.2 | - | - | 169.8 | - | - | 171.5 |
| 4 | 1.92 | dd, 5.2/5.2 | 47.8 | 2.64 | m | 46.3 | 1.92 | dd, 5.2/5.2 | 48.0 | 1.97 | dd, 5.0/5.0 | 52.4 | - | - | 79.2 |
| 5 | - | - | 37.2 | - | - | 36.9 | - | - | 37.2 | - | - | 37.2 | - | - | 42.9 |
| 6 | 2.59 2.02 | d, 17.5 d, 17.5 | 48.5 | 2.03 2.60 | d, 17.4 d, 17.4 | 48.0 | 2.59 2.02 | d, 17.6 d, 17.6 | 47.7 | 2.49 1.98 | d, 17.3 d, 17.3 | 48.0 | 2.13 2.16 | d, 18.0 d, 18.0 | 50.8 |
| 7 | 1.66 1.82 | m m | 26.8 | 1.62 1.88 | m m | 27.6 | 1.66 1.82 | m m | 27.1 | 1.69 1.80 | m m | 26.5 | 1.83 2.07 | m m | 34.6 |
| 8 | 1.63 | m | 37.1 | 1.60 | m | 36.0 | 1.63 | m | 37.2 | 1.68 1.61 | m m | 37.4 | 1.49 1.78 | m m | 32.7 |
| 9 | 3.82 | dd, 6.2/11.7 | 77.2 | 3.80 | m | 77.7 | 3.83 | dd, 6.3/11.7 | 77.7 | 3.83 | dd, 6.3/11.6 | 77.7 | 3.80 | ddd, 6.3/11.8/11.8 | 78.2 |
| 10 | 1.24 | d, 6.2 | 21.6 | 1.21 | d, 6.1 | 21.9 | 1.24 | d, 6.3 | 21.9 | 1.25 | d, 6.3 | 21.9 | 1.24 | d, 6.3 | 21.9 |
| 11 | 4.32 4.16 | dd, 17.8/1.8 dd, 17.8/1.8 | 64.9 | - | - | 160.1 | 4.33 4.16 | m/m | 64.9 | 2.05 | d, 1.2 | 24.9 | 2.04 | d, 1.0 | 21.7 |
| 12 | 1.02 | s | 28.4 | 1.01 | s | 28.4 | 1.02 | s | 28.6 | 1.02 | s | 28.7 | 1.01 | s | 24.3 |
| 13 | 1.12 | s | 27.5 | 1.12 | s | 27.5 | 1.11 | s | 27.5 | 1.10 | s | 27.4 | 1.09 | s | 23.7 |
| 1' | 4.31 | d, 7.9 | 103.8 | 4.30 | d, 7.9 | 104.1 | 4.32 | d, 7.9 | 103.8 | 4.32 | d, 7.8 | 104.0 | 4.31 | d, 7.9 | 104.0 |
| 2' | 3.15 | dd, 7.9/9.0 | 75.0 | 3.13 | dd, 7.9/8.9 | 75.1 | 3.15 | dd, 7.9/9.0 | 75.1 | 3.16 | dd, 7.8/9.0 | 75.2 | 3.14 | dd, 7.9/8.9 | 75.1 |
| 3' | 3.33 | dd, 9.0/9.0 | 77.9 | 3.35 | dd, 8.9/8.9 | 78.0 | 3.33 | m | 77.8 | 3.34 | dd, 9.0/9.0 | 78.0 | 3.33 | dd, 8.9/8.9 | 77.8 |
| 4' | 3.27 | dd, 9.0/9.0 | 71.4 | 3.27 | dd, 8.9/8.9 | 71.5 | 3.33 | m | 71.4 | 3.33 | dd, 9.0/9.0 | 71.5 | 3.27 | dd, 8.9/8.9 | 71.4 |
| 5' | 3.25 | m | 77.6 | 3.25 | m | 77.6 | 3.45 | m | 77.0 | 3.44 | m | 76.9 | 3.24 | m | 77.7 |
| 6' | 3.85 3.65 | dd, 11.8/2.2 dd, 11.8/5.5 | 62.5 | 3.84 3.66 | dd, 2.0/12.3 dd, 5.0/12.3 | 62.6 | 4.11 3.78 | dd, 11.7/2.0 dd, 11.7/5.7 | 69.6 | 4.11 3.79 | dd, 11.6/1.6 dd, 11.6/5.9 | 69.7 | 3.85 3.65 | d, 11.7 dd, 4.5/11.7 | 62.3 |
| 1'' | | | | | | | 4.40 | d, 7.9 | 104.6 | 4.40 | d, 7.8 | 104.8 | | | |
| 2'' | | | | | | | 3.21 | dd, 7.9/9.0 | 74.9 | 3.21 | dd, 7.8/9.0 | 75.0 | | | |
| 3'' | | | | | | | 3.34 | dd, 9.0/9.0 | 77.9 | 3.34 | dd, 9.0/9.0 | 77.9 | | | |
| 4'' | | | | | | | 3.28 | dd, 9.0/9.0 | 71.4 | 3.28 | dd, 9.0/9.0 | 71.5 | | | |
| 5'' | | | | | | | 3.26 | m | 77.9 | 3.26 | ddd, 9.0/5.4/1.8 | 78.0 | | | |
| 6'' | | | | | | | 3.87 3.66 | dd, 11.9/2.0 dd, 11.9/5.2 | 62.5 | 3.86 3.66 | dd, 11.6/1.8 dd, 11.6/5.4 | 62.7 | | | |

s, singlet; s br, broad singlet; d, doublet; dd, doublet of doublet; m, multiplet

DOI: https://doi.org/10.7554/eLife.37093.031

compounds. Column temperature was set to 42°C. As mobile phases, we used: A, 0.05% HCOOH, 0.1% ACN in $H_2O$ and B, MeOH, the composition of which was optimized for an efficient separation of blumenol-type compounds within a short run time. We included in the method a cleaning segment at 100% B and an equilibration segment allowing for reproducible results across large samples sets. The gradient program was as follows: 0–1 min, 10% B; 1–1.2 min, 10–35% B; 1.2–5 min, 35–50% B; 5–5.5 min, 50–100% B; 5.5–6.5 min, 100% B; 6.5–6.6 min, 100–10% B and 6.6–7.6 min, 10% B. The flow rate was set to 500 μL $min^{-1}$. Analysis was performed on a Bruker Elite EvoQ triple quadrupole MS equipped with a HESI (heated electrospray ionization) ion source. Source parameters were as follows: spray voltage (+), 4500V; spray voltage (-), 4500V; cone temperature, 350°C; cone gas flow, 35; heated probe temperature, 300°C; probe gas flow, 55 and nebulizer gas flow, 60. Samples were analyzed in multiple-reaction-monitoring (MRM) mode; the settings are described in *Table 2*.

## Method for targeted blumenol analysis in *N. attenuata*

The compound list was limited to the AMF-indicative markers in *N. attenuata*, Compound 1 and 2, the not AMF-indicative Compound 6 and the internal standard ($D_6$-ABA). Accordingly, the gradient

**Table 2.** MRM-settings used for targeted blumenol analysis.

| Nr. | Compound name | RT | Q1 [m/z]*, † | Q3 [m/z] ‡, § (CE [V]) |
|---|---|---|---|---|
| 1 | 11-hydroxyblumenol C-Glc¶, ** | 2.82 | +389.22 | 227.16 (-2.5), **209.15 (-7.5)**, 191.14 (-12.5), 163.10 (-15), 149.10 (-17.5) |
| 2 | 11-carboxyblumenol C-Glc¶, ** | 3.22 | +403.22 | 241.16 (-2.5), 223.15 (-7.5), 177.10 (-15), 195.14 (-12.5) |
| | | | +241.16 # | 223.15 (-5), 177.10 (-15), **195.14 (-10)** |
| 3 | 11-hydroxyblumenol C-Glc-Glc ¶, ** | 2.5 | +551.27 | 389.22 (-2.5), 227.16 (-7.5), **209.15 (-10)**, 191.14 (-15), 149.10 (-20) |
| 4 | Blumenol C – Glc-Glc ¶, ** | 3.47 | +535.27 | 373.22 (-2.5), **211.00 (-10)**, 193.10 (-17.5), 135.00 (-22.5), 109.00 (-22.5) |
| 5 | Blumenol C - Glc ¶, †† | 4.18 | +373.22 | **211.20 (-6)**, 193.16 (-9), 175.10 (-15), 135.12 (-16), 109.10 (-20) |
| 6 | Blumenol A - Glc¶, †† | 2.51 | - 385.20 | **153.10 (14)** |
| | | | +387.20 | 225.15 (-5), 207.14 (-8), 149.10 (-18), 135.12 (-16), 123.08 (-23) |
| 7 | Blumenol B - Glc¶, ** | 2.5 | +389.22 | 227.16 (-5), **209.15 (-7.5)**, 191.14 (-12.5), 153.10 (-17.5), 149.10 (-17.5) |
| 8 | Blumenol C – Glc-GlcU¶, ‡‡ | 3.25 | +549.27 | 373.22 (-2.5), **211.00 (-10)**, 193.10 (-17.5), 135.00 (-22.5), 109.00 (-22.5) |
| | | and 3.38 | | |
| 9 | 11-hydroxylumenol C – Glc-Rha‡‡ | 2.8 | +535.27 | 389.22 (-2.5), **227.16 (-7.5)**, 209.15 (-10), 191.14 (-15), 149.10 (-20) |
| 10 | Blumenol C – Glc-Rha¶, ‡‡ | 4.1 | +519.27 | 373.22 (-2.5), **211.00 (-10)**, 193.10 (-17.5), 135.00 (-22.5), 109.00 (-22.5) |
| 11 | Hydroxyblumenol C-Hex-Pen‡‡ | 2.5 | +521.27 | 389.22 (-2.5), 227.16 (-7.5), **209.15 (-10)**, 191.14 (-15), 149.10 (-20) |
| | $D_6$-ABA†† | 4.5 | - 269.17 | **159.00 (10)** |

RT: retention time

CE: collision energy

Glc: glucose

GlcU: glucuronic acid

Rha: rhamnose

Hex: hexose

Pen: pentose

*Resolution: 0.7

†[M + H]+ or [M-H]- if not stated differently

‡Resolution: 2

§ Quantifiers are depicted in bold

# [M + H-Glc]+

¶ Verified by high resolution MS

**Verified by NMR

††Optimized with commercial available standards

‡‡Transitions predicted based on structural similar compounds and literature information

DOI: https://doi.org/10.7554/eLife.37093.032

program was adjusted as follows: 0–1 min, 10% B; 1–1.2 min, 10–35% B; 1.2–3 min, 35–42% B; 3–3.4 min, 42–100% B; 3.4–4.4 min, 100% B; 4.4–4.5 min, 100–10% B and 4.5–5.5 min, 10% B. The MRM settings are given in *Table 3*.

## Determination of the AMF colonization rate

To determine the fungal colonization rates and mycorrhizal structures, root samples were stained and analyzed by microscopy. For WGA-Alexa Fluor 488 staining, roots were first washed with distilled water and then soaked in 50% (v/v) ethanol overnight. Roots were then boiled in a 10% (w/v) KOH solution for 10 min. After rinsing with water, the roots were boiled in 0.1 M HCl solution for 5 min. After rinsing with water and subsequently with 1x phosphate-buffered saline solution, roots were stained in 1x phosphate-buffered saline buffer containing 0.2 mg mL$^{-1}$ WGA-Alexa Fluor 488 overnight in the dark. Zeiss confocal microscopy (LSM 510 META) was used to detect the WGA-Alexa Fluor 488 (excitation/emission maxima at approximately 495/519 nm) signal. Trypan blue staining was performed as described by *Brundrett et al. (1984)* to visualize mycorrhizal structures. For the counting of mycorrhizal colonization, 15 root fragments, each about 1 cm long, were stained with either trypan blue or WGA-488 followed by slide mounting. More than 150 view fields per slide were surveyed with 20x object magnification and classified into five groups: no colonization, only hyphae (H), hyphae with arbuscules (H + A), hyphae with vesicles (V + H), and hyphae with arbuscules and vesicles (A + V + H). The proportions of each group were calculated by numbers of each group divided by total views.

For the molecular biological analysis of colonization rates, RNA was extracted from the roots using the RNeasy Plant Mini Kit (Qiagen) or NucleoSpin RNA Plant (Macherey-Nagel) according to the manufacturer's instructions and cDNA was synthesized by reverse transcription using the Prime-Script RT-qPCR Kit (TaKaRa). Quantitative (q)PCR was performed on a Stratagene Mx3005P qPCR machine using a SYBR Green containing reaction mix (Eurogentec, qPCR Core kit for SYBR Green I No ROX). We analyzed the *R. irregularis* specific housekeeping gene, *Ri-tub* (GenBank: EXX64097.1), as well as the transcripts of the AMF-induced plant marker genes *RAM1*, *Vapyrin*, *STR1* and *PT4*. The signal abundance was normalized to *NaIF-5a* (NCBI Reference Sequence: XP_019246749.1). The primer sequences are summarized in *Table 4*.

**Table 3.** MRM-settings for the analysis of selected blumenols in *N. attenuata*.

| Nr. | Compound name | RT | Q1 [m/z]*, † | Q3 [m/z]‡, § (CE [V]) |
|---|---|---|---|---|
| 1 | 11-hydroxyblumenol C-Glc¶, ** | 2.82 | +389.22 | 227.16 (-2.5), **209.15 (-7.5)**, 191.14 (-12.5), 163.10 (-15), 149.10 (-17.5) |
| 2 | 11-carboxyblumenol C-Glc¶, ** | 3.22 | +403.22 | 241.16 (-2.5), 223.15 (-7.5), 177.10 (-15), 195.14 (-12.5) |
| | | | +241.16# | 223.15 (-5), 177.10 (-15), **195.14 (-10)** |
| 6 | Blumenol A - Glc ¶, †† | 2.51 | - 385.20 | **153.10 (14)** |
| | | | +387.20 | 225.15 (-5), 207.14 (-8), 149.10 (-18), 135.12 (-16), 123.08 (-23) |
| | D$_6$-ABA†† | 4.0 | - 269.17 | **159.00 (10)** |

RT: retention time

CE: collision energy

Glc: glucose

Hex: hexose

Pen: pentose

*Resolution: 0.7

†[M + H]$^+$ or [M-H]$^-$ if not stated differently

‡Resolution: 2

§Quantifiers are depicted in bold

#[M + H-Glc]$^+$

¶Verified by high resolution MS

**Verified by NMR

††Optimized with commercial available standards

DOI: https://doi.org/10.7554/eLife.37093.033

**Table 4.** Sequences of primers used for qPCR-based analysis of AMF-colonization rates.

| Gene | Forward primer | Reversed primer |
| --- | --- | --- |
| NaIF-5a | GTCGGACGAAGAACACCATT | CACATCACAGTTGTGGGAGG |
| NaRAM1 | ACGGGGTCTATCGCTCCTT | GTGCACCAGTTGTAAGCCAC |
| NaVapyrin | GGTCCCAAGTGATTGGTTCAC | GACCTTCAAAGTCAACTGAGTCAA |
| NaSTR1 | TCAGGCTTCCACCTTCAATATCT | GACTCTCCGACGTTCTCCC |
| NaPT4 | GGGGCTCGTTTCAATGATTA | AACACGATCCGCCAAACAT |
| NaCCaMK | TTGGAGCTTTGTTCTGGTGGT | ATACTTGCCCCGTGTAGCG |
| NaNOPE1 | ACTTGATGCCATGTTTCAGAGC | TCCAATTCGCGATAAGCTGGT |
| Ri-TUB | TGTCCAACCGGTTTTAAAGT | AAAGCACGTTTGGCGTACAT |

DOI: https://doi.org/10.7554/eLife.37093.034

## Transcript analysis of the apocarotenoid pathway

The transcript analysis of the (apo)carotenoid pathway was conducted based on RNA-seq (Data Set 3) by using *N. attenuata* roots with or without *R. irregularis* inoculations. The data analysis methods are based on the previously published pipeline of *Ling et al. (2015)*. Representative values for transcripts abundances are TPM (Transcripts per kilobase of exon model per million mapped reads).

## Blumenol transport experiment

To analyze the root-to-shoot transfer potential of blumenols, we placed three *N. attenuata* seedlings, previously germinated on petri dishes with GB5 Agar for approximately 10 days, into 0.5 mL reaction tubes. The roots were placed into the tube, while the shoot projected out of the tube. The tubes were carefully covered with parafilm, which held the seedlings in place and isolated roots from shoots (see *Figure 6C*). The tubes were filled with tap water supplemented with 0.5% v/v plant extracts enriched in Compounds 1 or 2 (unknown concentration; purified fractions), or a commercial available standard of Compound 6 (25 ng $\mu L^{-1}$ end concentration; Roseoside; Wuhan ChemFaces Biochemical Co., Ltd.). Compound 1 or 2 were prepared from a mix of leaf tissues from different plant species (*M. truncatula, Z. mays, S. lycopersicum* and *N. attenuata*) by methanol extraction followed by purification by SPE (Chromabond HR-XC column) and HPLC (Agilent-HPLC 1100 series; Phenomenex Luna C18(2), 250 × 10 mm, 5 μm; equipped with a Foxy Jr. fraction collector). As a control, we used tap water supplemented with the respective amounts of MeOH. The seedlings were incubated for one day in a Percival climate chamber (16 hr of light at 28°C, and 8 hr of dark at 26°C). During sample collection, roots and shoots were separated and the roots were rinsed in water (to reduce the surface contamination with the incubation medium). While the shoots were analyzed separately, the roots of all seedlings from the same treatment were pooled. Sample extraction was conducted as described above.

## Inducible PDS silencing

For the temporal and spatial restriction of PDS gene silencing, we treated the petiole of the second oldest stem leaf of AMF-inoculated and non AMF-inoculated i-ir*PDS* and EV plants with a 100 μM dexamethasone-containing lanolin paste (1% v/v DMSO). The lanolin paste was prepared and applied as described by *Schäfer et al. (2013)*. The treatment started three weeks after potting and was conducted for three weeks. The lanolin paste was refreshed twice per week. On each plant the treated leaf and the adjacent, untreated leaves were harvested for analysis.

## QTL analysis

The field experiments for QTL analysis were conducted in 2017. Collected leaf samples were extracted as described with 80% MeOH spiked with $D_6$-ABA as internal standard and analyzed with the method described under '*Method for targeted blumenol analysis in N. attenuata*'. The peak areas for Compound 2 were normalized by the amount of extracted tissue and internal standard and log-transformed. Samples with missing genotype or phenotype information were removed. In total,

728 samples were used for QTL mapping analysis. For quantitative trait loci (QTL) mapping, we used the AZ-UT RIL population and data analysis described by *Zhou et al. (2017)*.

## Statistics

Statistical analysis of the data was performed with R version 3.0.3 (http://www.R-project.org/). The statistical methods used and the number of replicates are indicated in the figure legends.

## Acknowledgments

We thank the Brigham Young University for the use of their Lytle Preserve field station, Matthias Schöttner, Dechang Cao, Wenwu Zhou, Julia Cramer, Wibke Seibt and Eva Rothe for technical assistance, Danny Kessler, Andreas Schünzel, Andreas Weber, Jana Zitzmann from the glasshouse team for plant cultivation. This work was funded by the Max-Planck-Society, the ERC Advanced Grant (293926): ClockworkGreen and the Elsa Neumann Grant of Berlin, European Innovation Partnership Agri (276033540220041), Ministry of Consumer Protection, Food and Agriculture of the Federal Republic of Germany, Ministry for Science, Research and Culture of the State of Brandenburg, Thuringian Ministry of Infrastructure and Agriculture. Plant samples provided by the Harrison lab were generated with support from U.S. DOE # DE-SC0012460.

## Additional information

### Competing interests

Ian T Baldwin: Senior editor, eLife; European patent application EP 18 15 8922.7. Maria J Harrison: Reviewing editor, *eLife*. Ming Wang, Martin Schäfer, Dapeng Li, Rayko Halitschke, Erica McGale, Sven Heiling: European patent application EP 18 15 8922.7. The other authors declare that no competing interests exist.

### Funding

| Funder | Grant reference number | Author |
| --- | --- | --- |
| Max-Planck-Gesellschaft | Open-access funding | Ming Wang<br>Martin Schäfer<br>Dapeng Li<br>Rayko Halitschke<br>Chuanfu Dong<br>Erica McGale<br>Christian Paetz<br>Yuanyuan Song<br>Suhua Li<br>Junfu Dong<br>Sven Heiling<br>Karin Groten<br>Ian T Baldwin |
| European Innovation Partnership Agricultural Productivity and Sustainability | 276033540220041 | Philipp Franken<br>Michael Bitterlich |
| Ministry of Consumer Protection, Food and Agriculture of the Federal Republic of Germany | | Philipp Franken<br>Michael Bitterlich |
| Ministry of Science, Research and Culture of the State of Brandenburg | | Philipp Franken<br>Michael Bitterlich |
| Thuringian Ministry for Infrastructure and Agriculture | | Philipp Franken<br>Michael Bitterlich |
| Elsa-Neumann Scholarship | | Michael Bitterlich |
| U.S. Department of Energy | # DESC0012460 | Maria J Harrison |

| European Research Council | Advanced Grant ClockworkGreen (293926) | Ian T Baldwin |
| --- | --- | --- |

The funders had no role in study design, data collection and interpretation, or the decision to submit the work for publication.

### Author contributions
Ming Wang, Martin Schäfer, Conceptualization, Data curation, Formal analysis, Validation, Investigation, Visualization, Methodology, Writing—original draft, Writing—review and editing; Dapeng Li, Conceptualization, Data curation, Software, Formal analysis, Validation, Investigation, Visualization, Methodology, Writing—review and editing; Rayko Halitschke, Conceptualization, Data curation, Software, Formal analysis, Supervision, Validation, Investigation, Visualization, Methodology, Writing—original draft, Project administration, Writing—review and editing; Chuanfu Dong, Yuanyuan Song, Suhua Li, Junfu Dong, Investigation, Writing—review and editing; Erica McGale, Data curation, Software, Formal analysis, Investigation, Visualization, Writing—review and editing; Christian Paetz, Resources, Data curation, Formal analysis, Supervision, Investigation, Writing—review and editing; Sven Heiling, Validation, Investigation, Writing—review and editing; Karin Groten, Conceptualization, Supervision, Writing—review and editing; Philipp Franken, Maria J Harrison, Uta Paszkowski, Resources, Supervision, Writing—review and editing; Michael Bitterlich, Resources, Investigation, Writing—review and editing; Ian T Baldwin, Conceptualization, Resources, Supervision, Funding acquisition, Visualization, Writing—original draft, Project administration, Writing—review and editing

### Author ORCIDs
Martin Schäfer ![iD] http://orcid.org/0000-0002-4580-6337
Rayko Halitschke ![iD] http://orcid.org/0000-0002-1109-8782
Chuanfu Dong ![iD] http://orcid.org/0000-0003-3043-7257
Erica McGale ![iD] http://orcid.org/0000-0002-5996-4213
Philipp Franken ![iD] http://orcid.org/0000-0001-5710-4538
Michael Bitterlich ![iD] http://orcid.org/0000-0002-3562-7327
Maria J Harrison ![iD] http://orcid.org/0000-0001-8716-1875
Uta Paszkowski ![iD] https://orcid.org/0000-0002-7279-7632
Ian T Baldwin ![iD] http://orcid.org/0000-0001-5371-2974

### Decision letter and Author response
Decision letter https://doi.org/10.7554/eLife.37093.038
Author response https://doi.org/10.7554/eLife.37093.039

## Additional files
### Supplementary files
• Source code 1. Source code for QTL analysis.
DOI: https://doi.org/10.7554/eLife.37093.035
• Transparent reporting form
DOI: https://doi.org/10.7554/eLife.37093.036

### Data availability
All data generated or analysed during this study are included in the manuscript and supporting files. Source data files have been provided for all figures.

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
