## [Decision Letter]

Thank you for submitting your article "Blumenols as effective shoot markers for root symbiosis with arbuscular mycorrhizal fungi" for consideration by *eLife*. Your article has been reviewed by two peer reviewers, and the evaluation has been overseen by Detlef Weigel as the Reviewing and Senior Editor. The following individual involved in review of your submission has agreed to reveal their identity: Martin Parniske (Reviewer #1).

The reviewers and the editor agreed that this is potentially important work, showing that a relatively simple marker can be exploited to detect AMF colonization in the field. However, there was also agreement that you need to be clearer what the limitations of the work are (such as the apparent lag between colonization and blumenol accumulation) and that you need to give more credit to what has been possible before. In addition, we require the following essential revisions:

• Measure colonization and blumenol levels and directly compare both over a time course.

• Assess the correlation between colonisation and blumenol levels within at least a small number of RILs, to determine whether blumenol can mirror arbuscule abundance in the root at a very high level of resolution.

• Document that you can unequivocally identify blumenol C glucoside and not byzantionoside B. The two compounds are stereochemically different but carry the same molecular weight.

"Good-to-have" revisions:

We encourage you to look at a second CCaMK silenced line, if at all possible, and to ask how AMF colonization levels, as read out by blumenol, are reflected in plant performance in the field.

The full reviews are attached for your information.

*Reviewer #1:*

The observation that secondary plant compounds that accumulate in the leaves of all investigated plant species in response to AM colonisation is very exciting because of its novelty and its potential as a marker for the above-ground analysis of below-ground AM colonization without uprooting.

The authors observed that blumenols (and derivatives) specifically accumulate in the leaves of AM colonized plants. This phenomenon was consistently observed in all tested vascular plants species ranging from monocots such as rice to dicots such as tomato and legumes. They also show that this accumulation is specific to AM and is not induced by other tested biotic or abiotic stimuli. This in itself is already a very interesting observation. They obtained data from a cleverly designed "localized RNAi" assay the results of which, in combination with other analyses strongly suggest that specific AM-induced blumenols are transported from the AM colonized roots to the leaves.

The authors suggest to use blumenol levels in the leaves to screen for novel mutants/genotypes defective in AM. They present a convincing example in which they use blumenol levels in the leaves of Nicotiana RILs to map a genetic locus correlated with quantitative AM colonization differences.

Overall the work appears solid and the experiments are convincingly designed. The main claims of the authors are well supported by the data provided.

Major request: Unfortunately, the specific blumenols accumulate in the leaves only relatively late during the colonization (no signal at 4 weeks after inoculation; weak at 5 weeks; solid and screen-compatible levels only from 6 weeks onwards). This is not unexpected because blumenols were previously observed to increase over time in roots during AM development. However, depending on the cultivation and inoculation system, the colonization of the root typically happens weeks earlier. This delayed detection of the colonization somewhat limits the resolution possible with that setup. For example, slow colonizers that suffer from a slightly delayed colonization that may be detectable by microscopical analysis, may have caught up at such a late time point. To assist the evaluation of this assay by potential users, the authors should also present the data that indicate its limitations and determine the relationship between AM age, colonization level and blumenol levels. Colonization and blumenol levels should be measured and directly compared over a time course. As it stands, colonization alone was tested over a time course, but correlation between blumenol and colonization are only presented for one late timepoint.

*Reviewer #2:*

Summary:

1) The authors summarize work which provides evidence for the existence of markers accumulating in leaves of mycorrhizal host plants indicative for root colonization by AMF. This compounds include blumenols which have previously been suggested to compose the yellow pigment which accumulates in mycorrhizal roots. Blumenols are likely to be transported from roots to leaves in good correlation with mycorrhizal colonization and marker gene expression at the root level. Moreover, a high-throughput screening of blumenol levels in a RIL population provides a proof-of-principle that blumenols can be used as markers to identify gene loci involved in the AM symbiosis.

Significance and novelty:

The identification of leaf marker metabolites for root colonization by AMF in a broad range of AM symbiotic associations is timely and their use in HTP screening for root colonization is a promising tool to identify high yielding varieties exhibiting high root mycorrhizal colonization under adverse field conditions.

Abstract:

2) The authors report in the Abstract that foliar markers are not restricted to particular mycorrhizal species. It should be specified whether the foliar markers are metabolites other than blumenols or whether blumenols are addressed explicitly. Moreover, should "mycorrhizal species" comprise mycorrhizal host species, i.e. plants colonized by any mycorrhizal fungus, or specifically by AMF? The authors should use these terms more precisely according to their scientific definitions, because, alternatively, mycorrhizal species could also include AMF or other mycorrhizal fungi, since a mycorrhizal species, i.e. a species which is part of a mycorrhiza (root colonized by mycorrhizal fungus), could by definition be a plant or a fungus.

*eLife* Digest:

3) In their *eLife* Digest draft, the authors state that AMF symbioses are usually studied destructively and the analysis of mycorrhizal roots is time consuming. This is in essence true but it misses work from other groups which have shown that specific compounds accumulate in leaves of mycorrhizal plants, such as miRNA or secondary metabolites (such as e.g. terpenoids), or that leaf elemental profiles alter in response to root colonization. It would be interesting to know whether the authors have addressed these known physiological changes in their plants (leaves) and whether they have identified said compounds in their MS survey as well. This would show whether published results can be reproduced in *N. attenuata*.

4) According to Matsunami et al. (2010) blumenol C glucoside and byzantionoside B are stereochemically different but carry the same molecular weight. Have the authors considered this issue?

5) The authors experimentally interfere with carotenoid biosynthesis in leaves and conclude from unaltered blumenol content and the proven capacity of seedlings to transport blumenol from root to shoot that blumenol in the leaves originates from mycorrhizal roots. Although lacking final proof, e.g. through the application of labeled blumenols to the roots or through grafting with "blumenol mutants", the data resulting from independent approaches provides sufficient support for this hypothesis.

6) The authors claim that 728 plants can´t be realistically analyzed within two weeks. It is unclear which parameters were used for their calculations. For example, do they base their claim on the work capacity of one person?

Introduction:

7) In the Introduction it is stated that no general AMF-specific metabolites have been found previously. This statement somehow stays in contradiction with what has been shown in Adolfsson et al. (2017) and in Gerlach et al. (2015; cited in Adolfsson et al. 2017). Adolfsson et al. (2017) even show that genes involved in secondary metabolite biosynthesis are upregulated by AM colonization of roots. It is unclear to me why the authors believe that blumenol C glycoside is the first AM-specific compound found in leaves. Clarification is needed.

8) It is unclear in which work it has been shown that only blumenol C-based aglycone are positively correlated with mycorrhizal colonization (see Introduction, third paragraph). Moreover, I´m wondering whether it would be informative for a broad readership that blumenol is not the only apocarotenoid which accumulates in mycorrhizal roots (like e.g. mycorradicin possibly accumulating in the "yellow pigment" which is characteristic for AM roots). Despite my critical remarks I fully agree that a reliable leaf marker for efficient AM colonization would accelerate and facilitate breeding for beneficial AM symbiosis.

Results:

9) In Groten et al. (2015) three independent lines silenced in CCaMK were described. In the current work one of these lines was studied in detail. Why did the authors only study one line and did they confirm that the silencing effect was maintained in this line throughout their study? This is not apparent in the entire set of their results. The use of two independently transformed lines would probably have resulted in more robust results (e.g. with respect to Compound 1).

10) In Figure 2 untargeted metabolomics resulted in a number of AMF-specific hits. Wouldn´t it be interesting to know whether published work on compounds which accumulate in leaves upon mycorrhization also accumulate in *N. attenuata*? Moreover, blumenols and mycorradicin likely compose the yellow pigment in mycorrhizal roots. Could the authors detect mycorradicin or derivatives in the leaves of mycorrhizal plants? As I´m not a specialist in metabolomics, I don´t know, though, whether the used MS technology is sensitive enough to accurately identify these compounds.

11) As described in the Materials and methods, the field test described from line 20 onwards was performed with control and silenced plants alone or in combination in pots. It is likely that the AM symbiosis could establish at least to some degree when control and silenced lines were grown in combination. Did the authors investigate the degree of colonization by AMF in the roots of these plants and could reduced colonization in silenced plants explain why compound 2 provided a better quality marker than compound 1? Unfortunately data on growth phenotypes of field-grown as well as "lab-grown" plants is not provided to evaluate the impact of root interactions with AMF and other microorganisms on growth performance.

12) The authors denote the identified blumenol C-derivatives "AMF-indicative metabolites" and they show that the effect is not AMF taxa-specific. How can they be sure that other fungi colonizing plant roots and spanning the spectrum from parasitic over commensalistic to mutualistic interactions don´t affect blumenol biosynthesis in roots and its transport to the shoot?

13) The finding that blumenol derivatives can be used as a screening tool for forward genetics approaches is highly interesting and deserves great attention. In a HTP approach based on levels of compounds 1 and 2, the authors identify a locus which harbors the NOPE1 orthologue. They, however, do not provide evidence for differential expression of this gene in accessions UT and AZ which differ in their response to AMF. Since Figure 7 presents key data supposed to provide the proof of principle for the successful use of HTP screening towards breeding of crops improved in AM colonisation, this link between blumenol derivative levels, AMF colonization and expression of marker genes (Figure 7—figure supplement 1) should in my view include the NaNOPE1 gene. Moreover, physiological data revealing whether indicative blumenol level, marker gene expression and AMF colonization correlate with plant performance in the field would be highly interesting for agricultural applications and should be provided in this work if available.

Discussion:

14) The authors should evaluate whether the literature on mycorrhizal secondary metabolites detected in leaves doesn´t provide alternative candidates for HTP screening and whether these papers deserve being cited in the submitted work.

15) The addition of references linking to the transport of root-derived compounds to the shoot in mycorrhizal plants seems to be appropriate also in the Discussion (see also 7, 8, 14).

16) Yellow pigment accumulates predominantly in old and senescent arbscules. Blumenol levels in leaves correlate with colonization, but colonization per se is not a good marker for growth performance of mycorrhizal plants due to functional diversity in AMF-host plant interactions (see e.g. Smith et al., 2003 and 2004). Since the authors propose to use their tool to produce mycorrhiza-responsive and P-efficient high-yielding lines, it would be highly interesting, if not even required, to show more correlative data suggesting that blumenol levels in leaves correlate with plant growth performance at low phosphate conditions.

---

## [Author Response]

The reviewers and the editor agreed that this is potentially important work, showing that a relatively simple marker can be exploited to detect AMF colonization in the field. However, there was also agreement that you need to be clearer what the limitations of the work are (such as the apparent lag betweencolonization and blumenol accumulation) and that you need to give more credit to what has been possible before.

We agree that this aspect wasn’t sufficiently addressed and we hope that the revised text provides a more even-handed treatment of the limitations of the procedure in light of what was possible with previously available procedures. The specific changes are detailed in the responses to the respective reviewer comments below.

In addition, we require the following essential revisions:• Measure colonization and blumenol levels and directly compare both over a time course.

We agree that such a comparison would be useful and have added new data (Figure 3—figure supplement 2) to address this point.

• Assess the correlation between colonisation and blumenol levels within at least a small number of RILs, to determine whether blumenol can mirror arbuscule abundance in the root at a very high level of resolution.

We agree that this data would be nice to have, and it’s a major experimental objective of research planned for the 2019/2020 field seasons, when we will plant out a much more powerful MAGIC population into our Arizona field sites. However (for obvious manpower reasons), we did not harvest the roots, and conduct microscopic and transcriptional analyses of the 728 plants that were planted at the field sites in 2017. Nevertheless, we think that the robust correlation between the colonization and the blumenol levels in the RILs is well supported by the following data presented in the revised manuscript: 1) the comparison of all relevant parameters in the parental lines of RILs (Figure 7, Figure 7—figure supplement 1), which are consistent with the validation using: 2) the irCCaMK plants (Figure 3, Figure 3—figure supplement 3) under glasshouse and field condition; 3) the kinetic assay which directly compares the development of root colonization with the accumulation of the indicative compounds in roots and systemic leaves (Figure 3, Figure 3—figure supplement 2); and 4) the inoculum gradient test which establishes a direct correlation between experimentally titrated root colonization and the accumulation of the indicative compounds in leaves (Figure 3).

• Document that you can unequivocally identify blumenol C glucoside and not byzantionoside B. The two compounds are stereochemically different but carry the same molecular weight.

Thank you for pointing out the similarity of the two compounds which cannot be distinguished with our LCMS analysis. We carefully analyzed our NMR data and compared the results with previously published reference data^[1]^. Blumenol C glucoside and byzantionoside B differ only in the configuration of position C-9; blumenol C glucoside is (9*S*)-configured whereas byzantionoside B has the (9*R*)-configuration. Characteristic ^13^C-chemical shift differences can thus be found for the positions C-9, C-10 and C-1’. In byzantionoside, C-9 and C-10 were reported to have chemical shifts of δ_C_ 75.7 and δ_C_ 19.9, respectively. In contrast, the chemical shifts for the same positions in blumenol C glucoside were reported to be lowfield shifted to δ_C_ 77.7 and δ_C_ 22.0, respectively. Experimental chemical shifts of C-9 for the compounds identified in this publication were in the range from δ_C_ 77.2 to δ_C_ 78.2, and for C-10 in the range from δ_C_ 21.6 to δ_C_ 21.9, respectively. C-1’ of byzantionoside was reported to have a chemical shift of δ_C_ 102.3, while for blumenol C glucoside the chemical shift was δ_C_ 104.1. The experimental chemical shifts for C-1’ of the compounds of this publication are in the range from δ_C_ 103.8 to δ_C_ 104.1. Hence the ^13^C-chemical shift data are completely consistent with the structures being blumenol C glucosides rather than byzantionoside B.

More characteristic differences can be found in the ^1^H chemical shift data. The methylene shifts for H-7 of byzantionoside were reported to have chemical shifts of δ_H_ 1.50 and δ_H_ 1.98 while for blumenol C glucoside, the same position showed chemical shifts of δ_H_ 1.67 and δ_H_ 1.81. Experimental ^1^H chemical shifts for H-7 of the compounds 1-4 of this publication were found in the range of δ_H_ 1.62 to δ_H_ 1.69 and δ_H_ 1.80 to δ_H_ 1.88, respectively.

Consequently, the data clearly establish the structures to be blumenol C derivatives and not byzantionosides.

^[1]^Matsunami, Otsuka and Takeda, 2010.

"Good-to-have" revisions:We encourage you to look at a second CCaMK silenced line, if at all possible, and to ask how AMF colonization levels, as read out by blumenol, are reflected in plant performance in the field.

We agree that this information would provide desired robustness to our inferences and in the revision we have added data for an independently transformed irCCaMK line, A-09-1208-6 (to complement the data from line A-09-1212-1 which was presented in the first submission; see Figure 3—figure supplement 3; subsection “Plant material and AMF inoculation”). This irCCaMK line has already been well-characterized by Groten et al. (2015), including growth and fitness parameters. Adding the effect of AMF colonization on plant performance is clearly out of the scope of the current manuscript as it would require another 1-2 years of work (see comments below).

The full reviews are attached for your information.Reviewer #1:[…] Major request: Unfortunately, the specific blumenols accumulate in the leaves only relatively late during the colonization (no signal at 4 weeks after inoculation; weak at 5 weeks; solid and screen-compatible levels only from 6 weeks onwards). This is not unexpected because blumenols were previously observed to increase over time in roots during AM development. However, depending on the cultivation and inoculation system, the colonization of the root typically happens weeks earlier. This delayed detection of the colonization somewhat limits the resolution possible with that setup. For example, slow colonizers that suffer from a slightly delayed colonization that may be detectable by microscopical analysis, may have caught up at such a late time point. To assist the evaluation of this assay by potential users, the authors should also present the data that indicate its limitations and determine the relationship between AM age, colonization level and blumenol levels. Colonization and blumenol levels should be measured and directly compared over a time course. As it stands, colonization alone was tested over a time course, but correlation between blumenol and colonization are only presented for one late time point.

We thank the reviewer for this important comment. The changes are clearly observable already at the earlier time-points (Figure 3—figure supplement 2B-C, insertions), but the strong increases in accumulation of blumenols in the later stages in root and leaves visually mask the effect in the earlier stages, when the same y-axis is used to present the data. For example, in root tissues, good signals of Compound 1 were observed at 2 wpi, and already at 3 wpi compound 1 and 2 in leaves, signals which also mirrored root colonization (Figure 3, Figure 3—figure supplement 2). We added this information to the text (subsection “Hydroxy- and carboxyblumenol C-glucoside levels in leaves positively correlate with root colonizations”, second paragraph; subsection “Systemic AMF-mediated metabolite changes”, last paragraph) and address the differences between the available methods in the Discussion (see the aforementioned paragraph; subsection “AMF-indicative blumenols as tool for research and plant breeding”). Additionally, we added a figure that allows the comparison of blumenol levels and colonization rate over a time course (Figure 3—figure supplement 2).

Reviewer #2:[…] Abstract:2) The authors report in the Abstract that foliar markers are not restricted to particular mycorrhizal species. It should be specified whether the foliar markers are metabolites other than blumenols or whether blumenols are addressed explicitly. Moreover, should "mycorrhizal species" comprise mycorrhizal host species, i.e. plants colonized by any mycorrhizal fungus, or specifically by AMF? The authors should use these terms more precisely according to their scientific definitions, because, alternatively, mycorrhizal species could also include AMF or other mycorrhizal fungi, since a mycorrhizal species, i.e. a species which is part of a mycorrhiza (root colonized by mycorrhizal fungus), could by definition be a plant or a fungus.

We thank the reviewer for this comment and tuned the Abstract to make this point more precise.

eLife Digest:3) In their eLife Digest draft, the authors state that AMF symbioses are usually studied destructively and the analysis of mycorrhizal roots is time consuming. This is in essence true but it misses work from other groups which have shown that specific compounds accumulate in leaves of mycorrhizal plants, such as miRNA or secondary metabolites (such as e.g. terpenoids), or that leaf elemental profiles alter in response to root colonization. It would be interesting to know whether the authors have addressed these known physiological changes in their plants (leaves) and whether they have identified said compounds in their MS survey as well. This would show whether published results can be reproduced in N. attenuata.

We did not analyze all other reported traits induced by AMF inoculation in *N. attenuata* such as terpenoids and miRNA. Previous reports show various alterations in the leaf metabolism; however these currently known compounds didn’t qualify as general marker since they are often plant species dependent and not specific to AMF-interactions. The authoritative review of this work by Schweiger and Müller (2015) came to the conclusion: “AM-mediated effects on the leaf metabolome are highly diverse, with a plethora of metabolite classes being specifically modified in numerous plant species across various taxa. Even within the more conserved primary metabolism, no common response patterns to AM were found.” In the presented investigation we focused on the blumenols as a result from an untargeted metabolomics screen, but we agree that it might be interesting for future work to expand the analysis to other compounds. In the *eLife* digest we only shortly address this point to be easily understood by mentioning the absence of known “widespread and specific AMF-induced responses”; we address this point in more detail in the first paragraph of the Introduction.

4) According to Matsunami et al. (2010) blumenol C glucoside and byzantionoside B are stereochemically different but carry the same molecular weight. Have the authors considered this issue?

We thank the reviewer for pointing out that this information was missing and added it accordingly (please see the response to editor’s point 3).

5) The authors experimentally interfere with carotenoid biosynthesis in leaves and conclude from unaltered blumenol content and the proven capacity of seedlings to transport blumenol from root to shoot that blumenol in the leaves originates from mycorrhizal roots. Although lacking final proof, e.g. through the application of labeled blumenols to the roots or through grafting with "blumenol mutants", the data resulting from independent approaches provides sufficient support for this hypothesis.

Thanks for this!

6) The authors claim that 728 plants can´t be realistically analyzed within two weeks. It is unclear which parameters were used for their calculations. For example, do they base their claim on the work capacity of one person?

The claim was based on the experience in our laboratory of what is possible to accomplish by one person working with *N. attenuata* plants. We clarified this point in the manuscript (“eLife digest” section, last paragraph).

Introduction:7) In the Introduction it is stated that no general AMF-specific metabolites have been found previously. This statement somehow stays in contradiction with what has been shown in Adolfsson et al. (2017) and in Gerlach et al. (2015; cited in Adolfsson et al. 2017). Adolfsson et al. (2017) even show that genes involved in secondary metabolite biosynthesis are upregulated by AM colonization of roots. It is unclear to me why the authors believe that blumenol C glycoside is the first AM-specific compound found in leaves. Clarification is needed.

As mentioned in the second half of the first paragraph of the Introduction, various metabolite responses have been reported (including the work of Adolfsson et al., 2017). While these changes might play important roles in the plant-AMF interaction, they are not suitable markers as they are not specific to AMF interactions or vary among different plant taxa. Additionally, most studies focus on a single plant species making it hard to evaluate if the reported changes also apply to other taxa.

8) It is unclear in which work it has been shown that only blumenol C-based aglycone are positively correlated with mycorrhizal colonization (see Introduction, third paragraph). Moreover, I´m wondering whether it would be informative for a broad readership that blumenol is not the only apocarotenoid which accumulates in mycorrhizal roots (like e.g. mycorradicin possibly accumulating in the "yellow pigment" which is characteristic for AM roots). Despite my critical remarks I fully agree that a reliable leaf marker for efficient AM colonization would accelerate and facilitate breeding for beneficial AM symbiosis.

To our knowledge all reported AMF-induced blumenol glucosides contained a blumenol C-based aglycon. Still, no information is available that directly excludes other unknown blumenol aglycons and it was not our intention to claim this. We wanted to mention the observation regarding the currently known AMF-induced blumenols, and we agree that the sentence might be misunderstood and have rephrased it for clarity (Introduction, third paragraph). In our untargeted metabolomics analysis, mycorradicin did not show up. Like many AMF-colonized roots, *N. attenuata* roots also accumulate a yellow color in AMF inoculated plants, and it’s possible that our extraction and quantification method is not suitable for this compound. We added a comment mentioning mycorradicin as potential candidate for further studies to the Discussion (subsection “Systemic AMF-mediated metabolite changes”, last paragraph).

Results:9) In Groten et al. (2015) three independent lines silenced in CCaMK were described. In the current work one of these lines was studied in detail. Why did the authors only study one line and did they confirm that the silencing effect was maintained in this line throughout their study? This is not apparent in the entire set of their results. The use of two independently transformed lines would probably have resulted in more robust results (e.g. with respect to Compound 1).

Please see the response to the editor’s point 4.

10) In Figure 2 untargeted metabolomics resulted in a number of AMF-specific hits. Wouldn´t it be interesting to know whether published work on compounds which accumulate in leaves upon mycorrhization also accumulate in N. attenuata? Moreover, blumenols and mycorradicin likely compose the yellow pigment in mycorrhizal roots. Could the authors detect mycorradicin or derivatives in the leaves of mycorrhizal plants? As I´m not a specialist in metabolomics, I don´t know, though, whether the used MS technology is sensitive enough to accurately identify these compounds.

Please see the response to reviewer 2’s point 3 and 8. The identified MS features are provided in Figure 2—figure supplement 1—source data 1.

11) As described in Materials and methods, the field test described from line 20 onwards was performed with control and silenced plants alone or in combination in pots. It is likely that the AM symbiosis could establish at least to some degree when control and silenced lines were grown in combination. Did the authors investigate the degree of colonization by AMF in the roots of these plants and could reduced colonization in silenced plants explain why compound 2 provided a better quality marker than compound 1? Unfortunately data on growth phenotypes of field-grown as well as "lab-grown" plants is not provided to evaluate the impact of root interactions with AMF and other microorganisms on growth performance.

We agree with the reviewer that this is an interesting point. irCCaMK plants in the co-cultured system have been intensively studied and the growth parameters under glasshouse and field conditions can be found in Groten et al. (2015) and Wang et al. (2017). Quantifying the effect of the AMF interaction on the plant performance is clearly out of the scope of this manuscript as it will require many additional years of field work to robustly understand. In general, despite years of optimization, the growth conditions required for AMF association in the glasshouse produce plants which are smaller and slower growing than those grown under field conditions. So a scientifically robust answer to this question is not a trivial endeavor. At the moment we are not sure about the reason for the lower quality of Compound 1 under field conditions, but suspect that it might be caused by other co-occurring compounds within the leaf extracts of field-grown plants which interfere with the detection.

12) The authors denote the identified blumenol C-derivatives "AMF-indicative metabolites" and they show that the effect is not AMF taxa-specific. How can they be sure that other fungi colonizing plant roots and spanning the spectrum from parasitic over commensalistic to mutualistic interactions don´t affect blumenol biosynthesis in roots and its transport to the shoot?

We agree with the reviewer that this can’t be completely ruled out, but based on the currently available experimental data, various AMFs are known to induce these compounds, while other tested biotic interaction partners failed to do so (see Figure 5 and Maier et al., 1997). Based on previous reports and our data, blumenol accumulations appear to be robustly associated with the formation of arbuscules, which is likely the reason for the specificity.

13) The finding that blumenol derivatives can be used as a screening tool for forward genetics approaches is highly interesting and deserves great attention. In a HTP approach based on levels of compounds 1 and 2, the authors identify a locus which harbors the NOPE1 orthologue. They, however, do not provide evidence for differential expression of this gene in accessions UT and AZ which differ in their response to AMF. Since Figure 7 presents key data supposed to provide the proof of principle for the successful use of HTP screening towards breeding of crops improved in AM colonisation, this link between blumenol derivative levels, AMF colonization and expression of marker genes (Figure 7—figure supplement 1) should in my view include the NaNOPE1 gene. Moreover, physiological data revealing whether indicative blumenol level, marker gene expression and AMF colonization correlate with plant performance in the field would be highly interesting for agricultural applications and should be provided in this work if available.

We added the expression data for the NOPE1 orthologue in the parent lines (Figure 7—figure supplement 1B, Table 4). Still, NOPE1 is just a potential candidate and the analysis of the QTL data will require extensive additional work, as mentioned in the manuscript. We agree that it would be interesting to correlate the blumenol levels to the respective growth performance data, but again this is clearly out of the scope of the current manuscript.

Discussion:14) The authors should evaluate whether the literature on mycorrhizal secondary metabolites detected in leaves doesn´t provide alternative candidates for HTP screening and whether these papers deserve being cited in the submitted work.

We agree that probably other compounds exist that would be suitable AMF markers. But we have spent months scouring the literature and, in agreement with the conclusion of Schweiger and Müller (2015), to the best of our knowledge no other markers with similar characteristics (AMF specific, responsive in various plant species, induced by various AMF) have been reported so far. More general information about the value of understanding AMF-induced changes in leaf metabolism is provided in the first paragraph of the Introduction.

15) The addition of references linking to the transport of root-derived compounds to the shoot in mycorrhizal plants seems to be appropriate also in the Discussion (see also 7, 8, 14).

We agree with the reviewer and added additional references to the Discussion.

16) Yellow pigment accumulates predominantly in old and senescent arbscules. Blumenol levels in leaves correlate with colonization, but colonization per se is not a good marker for growth performance of mycorrhizal plants due to functional diversity in AMF-host plant interactions (see e.g. Smith et al., 2003 and 2004). Since the authors propose to use their tool to produce mycorrhiza-responsive and P-efficient high-yielding lines, it would be highly interesting, if not even required, to show more correlative data suggesting that blumenol levels in leaves correlate with plant growth performance at low phosphate conditions.

The plant-AMF interaction is highly complex and the effects on plant performance depends on various factors, also including other factors than just the colonization rate – still the colonization rate is one of them. The accumulation of AMF-induced blumenols is well correlated with the abundance of arbuscules – the core structure of AMF interactions. Therefore, we suspect it to be a suitable proxy for a functional AMF interaction. We hope that before Baldwin retires from the MPG (in 8 years), we will be able to provide a robust analysis of the effect of colonization on plant performance.